# Node Classification in the Heterophilic Regime via Diffusion-Jump GNNs

## Abstract

In the heterophilic regime (HR), vanilla GNNs learn latent spaces where nodes with different labels may have similar embeddings. As a result, the performance of node classification degrades significantly in this context. However, existing metrics for heterophily count local discontinuities instead of characterizing heterophily in a structural way. In the ideal (homophilic) regime, nodes belonging to the same community have the same label: most of the nodes are harmonic (their unknown labels result from averaging those of their neighbors given some labeled nodes). Harmonic solvers are natural minimizers of the Laplacian Dirichlet energy. Therefore, a homophilic network is more harmonic than any heterophilic version of the same network. In other words, heterophily can be seen as a "loss of harmonicity". In this paper, we define "structural heterophily" in terms of the ratio between the harmonicity of the network (Laplacian Dirichlet energy) and the harmonicity of its homophilic version (the so-called "ground" energy).

In this paper, we also propose a novel GNN model (Diffusion-Jump GNN) that bypasses structural heterophily by "jumping" through the network in order to relate distant homologs. However, instead of using hops as standard High-Order (HO) GNNs (MixHop) do, our jumps are rooted in a structural well-known metric: the diffusion distance. Given the diffusion distances matrix (DM), we explore different orders of distances wrt each node (closest node, second closest node, etc.) in parallel. Each parallel exploration defines a "jump" that masks the network: it is a new graph that feeds a vanilla GNN layer. Consequently, different GNNs attend to different slices of the DM. As a result, we allow distant homologs to have similar embeddings in (at least) one of the jumps. In addition, as the final embedding of each node depends on the concatenation of its parallel embeddings, we can capture the explainability of each jump via learnable coefficients.

Since computing the DM is the core of this method, our main contribution is that we learn both the diffusion distances and the "coefficients" of the edges associated with each jump, thus defining "learnable structural filters". In order to learn the DM, we exploit the fact that diffusion distances have a spectral interpretation. Instead of computing the eigenvectors of the Laplacian, we learn orthogonal approximations of the Fiedler vector solving a trace-ratio optimization problem while the prediction loss is minimized. This leads to an interplay between a Dirichlet loss, which captures low-frequency content, and a prediction loss which refines that content leading to empirical eigenfunctions. Finally, our experimental results show that we are very competitive with the SOTA both in homophilic and heterophilic datasets, even in large graphs.

## 1 Introduction

The success of Graph Neural Networks (GNNs) relies on their convolutional architecture Kipf & Welling (2017)Hamilton et al. (2017)Veličković et al. (2018). Their *aggregate and combine* mechanism provides a significant degree of expressiveness. Consider, for instance, the task of semi-supervised *node classification* Song et al. (2021). Given an input graph, where each node is attributed with a matrix of features, some of the nodes are labeled. The goal of node classification is to infer the unknown labels *by using the network structure* Zhu et al. (2020). GNNs usually work in two steps: (a) computing a new (latent) representation for each node and (b) using the representations

of nodes with known labels to infer those of the unlabeled nodes. Actually, the focus of GNNs is on computing the latent representations whereas the classification is performed by a subsequent multi-layer perceptron (MLP). In this regard, the latent representation of a given node is computed by aggregating the features of its neighbors and then combining this aggregation with its own features in a robust way (usually through a local MLP Hamilton et al. (2017)).

Under *homophily* (neighboring nodes tend to have the same label McPherson et al. (2001)), aggregating the features of neighboring nodes and combining them in robust representations leads to hash the latent representations of *homologs* (nodes of the same class) pretty close. For instance, in Fig. 1-Top (graph **A**), only the nodes in the bottleneck (belonging to the edges defining the cut between two communities) tend to mix heterogeneous features, thus being potentially misclassified. In the *heterophilic* regime (HR), however, linked nodes are likely from different classes (Fig. 1-Bottom, graph **B**), and the risk of misclassification is higher. Consequently, for a vanilla GNN, the *level of heterophily* is a proxy of the misclassification risk. For instance, *node homophily* $h_{node}$ is the fraction of nodes connected with homologs Pei et al. (2020b). However, *edge homophily* $h_{edge}$ is the fraction of edges that connect nodes that

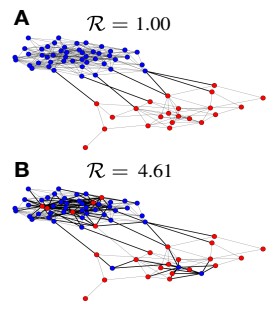

**A**    $\mathcal{R} = 1.00$

**B**    $\mathcal{R} = 4.61$

Figure 1: Structural heterophily.

have the same class label (i.e., intra-class edges) Zhu et al. (2020). Finally, *class homophily* $h_k$, where $k$ is a class-index, is the ratio between the sum of *restricted* degrees (count only neighbors of the same class) for nodes of class $k$ and the sum of degrees for the same nodes Lim et al. (2021).

In this paper, we propose **structural heterophily**, $\mathcal{R}$, a measure that is more complex than counting pairwise interactions but simpler than hypergraph-based measures Veldt et al. (2023). We leverage the combinatorial nature of graphs, i.e. **A** and **B** in Fig. 1 represent different ways of coloring the same structure, say $G = (V, E)$. From the point of view of spectral graph theory Chung (1997), the **A** coloring encodes the partition of $G$ minimizing the normalized cut Shi & Malik (2000), i.e. only nodes in the bottleneck are not linked with their homologs. Such a partition is approximated by the second non-trivial eigenvector of the graph Laplacian (the Fiedler vector) and the corresponding eigenvalue (the spectral gap) quantifies the so-called *ground energy* associated with $G$. Consequently this labeling the minimal structural heterophily $\mathcal{R} = 1$ (wlog, this also applies to graphs with $k > 1$ communities). As a result, any sub-optimal labeling wrt the normalized cut problem, e.g. **B**, is approximated by a Laplacian eigenvector associated with a larger eigenvalue: the structural heterophily for $G$ labeled with **B** is $\mathcal{R} = 4.61$ (almost five times more heterophilic than $G$ labeled with **A**). In the figure, high-energy edges are displayed in bold.

Structural heterophily can be interpreted as a *loss of harmonicity*. Harmonic structures are low-energy configurations where the label of a node is the average of those of its neighbors. If only some labels are known, as in semi-supervised learning, the "gaps" are naturally filled attending to maximize harmonicity. It is not surprising that harmonic solvers are natural minimizers of the Laplacian Dirichlet energy Doyle & Snell (2000)Grady (2006). This suggests that enforcing harmonicity under strong heterophily is a good idea: we will learn eigenvectors (called **empirical eigenfunctions**) which will be as much harmonic as possible while they react to heterophilic labelings.

It seems also reasonable to solve the heterophily issue by surfing through the graph to link distant homologs. As we will review in the Related-Work Section 6, our approach is aligned with **High-Order GNNs** Yan et al. (2022)Abboud et al. (2022) Song et al. (2023) Abu-El-Haija et al. (2019)Maurya et al. (2021)Maurya et al. (2022)Frasca et al. (2020). Most of these works are *hop-based*, i.e. they rely on powers of the transition matrix. As we show in Fig. 2-Top, these methods build a hop-hierarchy. However, such a hierarchy does not encode the dynamics of the random walks implicitly defined by the powers of the transition

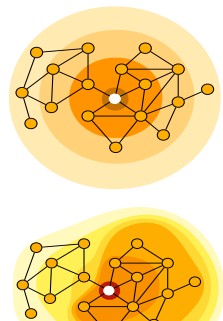

Figure 2: Hop-hierarchy (Top) vs Jump-hierarchy (Bottom). Diffusion distances contract the similarity space due to *structural forces*.

matter. For instance, a random walk rooted in the white node tends to visit its own community before leaving it: the so-called *escape probability* depends on the ground energy or spectral gap Meilă & Shi (2001) (see below and Appendixes B.3 and B.4). One way of designing a hierarchy that encodes the dynamics of random walks is to learn **diffusion distances** Nadler et al. (2005) (see below and Appendix B.2) which emphasizes the notion of a cluster. These distances have a spectral interpretation and they can be computed from our empirical eigenfunctions. The resulting hierarchy (Fig. 2-Bottom), whose level sets are called **jumps**, incorporates the dynamics of the random walks, thus relying on what we call *structural forces* (the inverses of the escape probabilities).

Summarizing, our two main contributions are a) a novel measure for heterophily called **structural heterophily**, and b) a novel **High-Order GNN architecture** inspired by this measure which relies on a fully learnable jump hierarchy. This implies both learning the empirical eigenfunctions and exploring many jumps in parallel. The final objective is to bypass the structure of the graph so that distant homologs have similar representations in the latent space. This process is called *homophiliation*.

## 2 HETEROPHILY AS THE LOSS OF HARMONICITY

**Node-classification under heterophily** can be posed as the following semi-supervised learning problem. Given an input graph, $G = (V, E)$ with adjacency matrix $\mathbf{A}$ and node features $\mathbf{X}$, there is a node subset $B \subset V$ whose labels $\ell(B)$ are known by the learner (border nodes). Similarly, the labels $\ell(U)$ of the remaining nodes, those in $U = V - B$, are hidden (interior nodes).

Given the graph Laplacian $\triangle := \mathbf{D} - \mathbf{A}$, where $\mathbf{D}$ is the diagonal degree matrix, and a regularizer (minimizer of $\mathbf{x}^T \triangle \mathbf{x} := \sum_{i \sim j} (\mathbf{x}_i - \mathbf{x}_j)^2$), we obtain $\ell^* = \arg \min_\ell \ell^T \triangle \ell$, where $\ell^*$ is the smoothest labeling of $V$ after propagating $\ell(B)$ to $\ell(U)$ through the edges of the graph. A Dirichlet solver ensures that the labeling $\ell^*$ is *harmonic* (the label of a given unknown node is the average of those of its neighbors) subject to the labeling of the border nodes $\ell(B)$ (see details in Appendixes A and C).

In the HR, two neighboring nodes rarely share their labels. As a result, $\ell^{*T} \triangle \ell^* \gg \mathbf{u}^T \triangle \mathbf{u}$, where $\mathbf{u}$ are the vectorized labels obtained by an alternative unsupervised learner (minimizer of the normalized-cut). The unsupervised learner typically assumes that the labels $\mathbf{u}$ are similar within each cluster (homophily). In other words, *heterophily can be posed in terms of how much harmonicity is lost wrt the homophilic assumption*.

The objective of a GNN is to learn a parametric function $f_\Theta(\mathbf{A}, \mathbf{X}, \ell(B))$ returning $\mathbf{H}$, a matrix (embedding) of latent representations (one row per node) so that the embeddings of either border nodes or hidden nodes with the same label are grouped together. However, $f_\Theta(.)$ does not necessarily minimize $c(\mathbf{H})^T \triangle c(\mathbf{H})$, where $c(.)$ contains the vectorized classification labels. We need to infer a *hidden graph* $G' = (V, E')$ where $c(\mathbf{H})^T \triangle_{G'} c(\mathbf{H})$ is minimized. Actually, the edges $E$ in the hidden graph should link nodes with the same label, even if they are in the same community or not.

**Structural Heterophily.** Given the above formulation, we may characterize heterophily in a structural way, namely *as the departure from a structural unsupervised grouping*. In particular, the ratio

$$\mathcal{R} = \frac{\ell^T \triangle \ell}{\mathbf{u}^T \triangle \mathbf{u}} \geq 1 \tag{1}$$

is close to the unit if the graph is homophilic (cuts in the structure mean discontinuities in the labeling). For $\mathcal{R} > 1$ the graph is heterophilic. The larger the ratio the larger the heterophily.

We use the example in Fig. 2-Bottom to illustrate how $\mathcal{R}$ works. We have two communities, $V = A \bigcup \bar{A}$ (left and right respectively). The *white node* belongs naturally to the right one $\bar{A}$, and this is what an unsupervised structural clustering detects: the Fiedler vector $\mathbf{u} = \arg \min_{\mathbf{x} \neq \mathbf{0}, \mathbf{x} \perp \mathbf{1}} \mathbf{x}^T \triangle \mathbf{x}$ has positive components ($\approx +1$) in $A$ and negative components ($\approx -1$) in $\bar{A}$.

The vector $\mathbf{u}$ is the smallest nontrivial eigenvector of $\triangle$ as well as the largest nontrivial eigenvector of the transition matrix $\mathbf{P} := \mathbf{D}^{-1} \mathbf{A}$, where $\mathbf{A}$ is the adjacency matrix. It has been argued that the top eigenvectors of $\mathbf{P}$ may be used to decompose the state space into metastable subspaces Huisinga et al. (2004). In other words, each of the two graph communities in Fig. 2 defines a metastable state from which a random walker tries to escape (Appendix B.4).

The average escape time is the inverse of the top nontrivial eigenvalue of the transition matrix $\mathbf{P}$, i.e. the inverse of the approximated spectral gap Matkowsky & Schuss (1981)Hänggi et al. (1990)Nadler

& Galun (2006). In our example, the spectral gap is very tiny so we can expect large escape times. In particular, the two states defined by the Fiedler vector are very compact (they have low variability). As a result, all the pairs of nodes $(i, j)$ inside each community have very similar **diffusion distances** $d(i, j)$ (defined in Appendix B.2) according to the structural forces characterizing each metastable state.

Consequently, the jump hierarchy defines a succession of unstable states $\mathbf{u}_1, \mathbf{u}_2, \ldots$ resulting from the expansion from $\bar{A}$: $\bar{A} \subseteq \bar{A}_1 \subseteq \bar{A}_2 \subseteq \ldots$. They are unstable because their Dirichlet energies $\mathbf{u}_k^T \triangle \mathbf{u}_k$ are greater than that of the unsupervised clustering (*ground energy*) $\mathbf{u}^T \triangle \mathbf{u}$.

Last, but by no means least, if we label the white node as belonging to $A$ instead of belonging to $\bar{A}$ (i.e., *we introduce heterophily*), we also increase the Dirichlet energy wrt the ground energy, i.e. $\ell^T \triangle \ell > \mathbf{u}^T \triangle \mathbf{u}$. Why? This is because the new Fiedler vector $\mathbf{u}_\ell$ leading to the labeling $\ell$ does no longer induce a sharp step function. This is consistent with the increase of the spectral gap and the reduction of the escape time.

Therefore, one useful interpretation of heterophily in structural terms (departure from the ground energy) is the fact that heterophily relaxes Dirichlet energies in such a way that it is possible to escape from a community in a few jumps and then find nodes with the same label in other communities. Therefore, paying attention to several jumps simultaneously increases the chance of aggregating homologs, thus solving the heterophily issue.

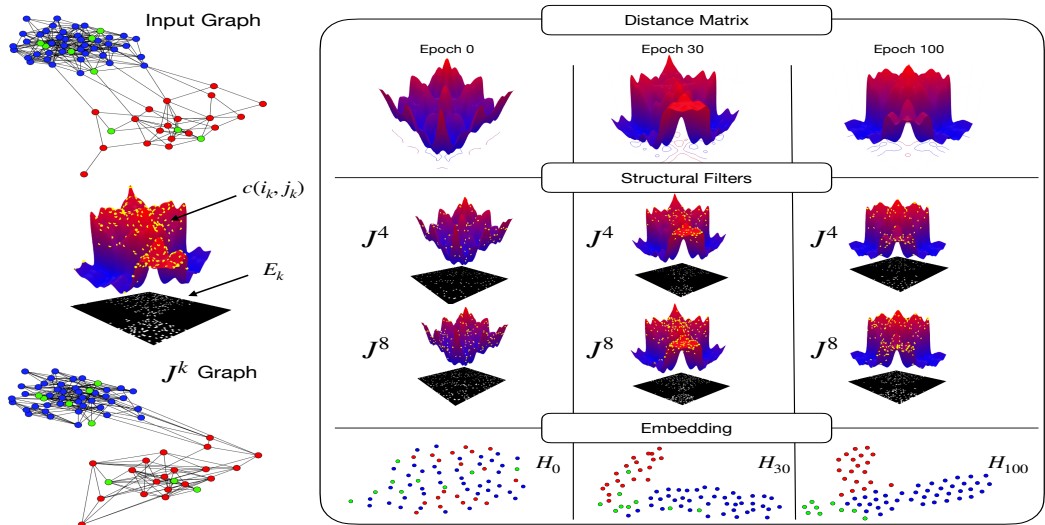

Figure 3: Homophiliation. Top-Left: a heterophilic graph. Center-Left: The current distance matrix $\mathcal{D}$ leads to a tridimensional weight distribution $c(i, j) = e^{-d(i,j)}$. Yellow points denote the support $E_k = \{(i_k, j_k)\}$ of the filter $\mathbf{J}^k$. The filter coefficients are given by the weights of the support $C_k = \{c(i_k, j_k)\}$. Bottom-Left: graph used for aggregation with this filter. In the right panel, we show the weight distribution (top), a couple of filters (middle), and the resulting homophiliation (bottom) for some epochs. In particular, we show $\mathbf{H}_0$, $\mathbf{H}_{30}$ and $\mathbf{H}_{100}$. In each epoch $e$, all the embeddings $\{\mathbf{H}_e^k\}$ contribute to identifying potential links between distant homologs. If any of these links is wrong the matrix of pairwise distances $\mathcal{D}$ is updated.

## 3 HOMOPHILIATION: LOSSES AND MODULES

**Homophiliation.** Our computational model for node-classification under heterophily cannot only rely on enforcing harmonic labelings, even when it reacts to the training labels. It must also transform the matrix of node features $\mathbf{X}$ into a piecewise-smooth embedding $\mathbf{H}$. The rows in $\mathbf{H}$ associated with nodes with the same label must be clustered together and these labels must be consistent with those of the border nodes $\ell(B)$. Such a process, i.e. the learning of the GNN $f_\Theta(\mathbf{A}, \mathbf{X}, \ell(B))$, results from

solving the following optimization problem:

$$
\begin{aligned}
\min \ & \mathcal{L} = \mathrm{Tr}[f_\theta(\mathbf{A})^T \triangle f_\theta(\mathbf{A})] + \mathcal{L}_c(\{\mathbf{J}^k\}, \mathbf{X}, \ell(B)) \\
\text{s.t.} \ & \mathbf{U}^T\mathbf{U} = \mathbf{I} \\
& \mathbf{U} = f_\theta(\mathbf{A}), \ \mathcal{D}(i,j) = \|\nabla\mathbf{U}_{ij}\| \ \text{ and } \ \mathbf{J}^k = \Pi^k \cdot \exp\left(-\mathcal{D}\right) \ ,
\end{aligned}
\tag{2}
$$

where we have an interplay between the *Dirichlet loss* $\mathrm{Tr}[f_\theta(\mathbf{A})^T \triangle f_\theta(\mathbf{A})]$ and the *classification loss* $\mathcal{L}_c$ as follows.

**Diffusion Pump.** Minimizing the structural heterophily so that $\mathcal{R} \approx 1$ (Eq. 1) in $G' = (V, E')$ implies learning Dirichlet energies close to the ground energy. However, in the heterophilic regime, we cannot minimize $c(\mathbf{H})^T \triangle_{G'} c(\mathbf{H})$ before discovering the optimal embedding $\mathbf{H}^*$. In the meanwhile, the Dirichlet formulation allows us to learn the smallest nontrivial eigenvectors of $\triangle$ as we do in the unsupervised setting (e.g. the Fiedler vector). These eigenvectors will be in the columns of $\mathbf{U}$, but they are parameterized as follows. The notation $\mathbf{U} = f_\theta(\mathbf{A})$, where $f_\theta$ is an MLP, goes beyond emphasizing the learnability of $\mathbf{U}$: they are the projections/transformations of the adjacency $\mathbf{A}$. In addition, they react to the classification loss during backpropagation (empirical eigenfunctions).

We learn the eigenvectors $\mathbf{U}$ because it is key to computing diffusion distances between the nodes. In the following, we will replace $d(i,j)$ by $\mathcal{D}(i,j)$ when we need to emphasize the matrix nature of the pairwise distances. Each pairwise distance $\mathcal{D}(i,j)$ comes from the norm of $\nabla\mathbf{U}_{ij} = \mathbf{U}_{i:} - \mathbf{U}_{j:}$ (row-difference). As we will detail in Section 4, $\|\nabla\mathbf{U}_{ij}\|$ approximates the diffusion distance between two nodes $i$ and $j$. Herein, we focus on the fact that nodes belonging to the same sub-structure (e.g. cluster or community) have similar distances. Back to Fig. 2, if $i$ and $j$ belong to the same community, two random walks placed in $i$ and $j$ have similar escape probabilities. Therefore, we build an anisotropic hierarchy of escape probabilities to characterize the respective reachability of any node from a given one. This latter hierarchy is built by specifying binary projection matrices $\Pi^k$ which select the pairs of distances that support the creation/update of each structural filter $\mathbf{J}^k$ (Section 4).

**Exploration by Parallel Jumping.** The diffusion pump triggers the creation of $K+1$ structural filters $\{\mathbf{J}^0, \mathbf{J}^1, \ldots, \mathbf{J}^K\}$ derived from their respective jumps $J^k$. Each filter $\mathbf{J}^k = (V, E_k, C_k)$ is the weight matrix of and edge-attributed graph with support $E_k = \{(i_k, j_k)\}$ and coefficients $c(i_k, j_k) = e^{-d(i_k, i_k)}$. We illustrate this process in Fig. 3 over a heterophilic graph. At any epoch, the optimizer creates a distance matrix $\mathcal{D}$ and weighs it: $C = \exp[-\mathcal{D}]$. The result is a weight distribution (middle-left). Each yellow point in the weight distribution belongs to the jump $J^k$. Consequently, the yellow points denote the edges of the filter support $E_k$ and they are projected in the adjacency matrix below the distribution. The coefficients $c(i_k, j_k)$ of the filter are the heights of the yellow points. Finally, the edges in the graph depicted below the distribution are exactly those of the filter support. Neighbor aggregation wrt this filter $\mathbf{J}^k\mathbf{X}$ exploits both that graph and these weights.

The right panel in Figure 3 shows the evolution of the weight distributions (top), some filters (center), and the status of the homophiliation process (bottom). The optimization process is initially dominated by the diffusion pump since random weight distributions are explored first. As a result, distant homologs can be potentially aggregated: we start to implicitly build the hidden graph $G' = (V, E')$. The probability of aggregating distant nodes is leveraged by the fact that, during the first epochs, most of the $K+1$ filters $\mathbf{J}^k$ have a random nature independently of $k$, the filter order. Escape probabilities are relaxed during this *exploration stage*.

**Classification/prediction Loss.** Each filter, $\mathbf{J}^k = (V, E_k, C_k)$ is a learnable graph that feeds a vanilla GNN $\sigma(\mathbf{J}^k\mathbf{X}\mathbf{W}^k)$ which generates the embedding $\mathbf{H}^k$. This embedding is weighted by a learnable parameter $\alpha_k$ and concatenated with the remaining embeddings to feed a classification layer. Therefore, as soon as the structural filters discover interesting bonds for minimizing $\mathcal{L}_c$, the weights $\mathbf{W}^k$ of all the GNNs, the filters' coefficients, and the distance matrix will become more and more stable. At some point in the optimization process, the Dirichlet loss will be stabilized and the exploration stage ends. Later on, the classification loss will refine the almost-homophilic global embedding $\mathbf{H}$. As a result, the embeddings of either border nodes or hidden nodes with the same label are grouped together in the latent space (for instance, see the column of Epoch 100 in Figure 3).

## 4 METHODOLOGICAL DETAILS

### 4.1 NETWORK ARCHITECTURE

DIFFUSION-JUMP GNNs are neural networks $f_\Theta(\mathbf{A}, \mathbf{X}, \ell(B))$ resulting from the optimization problem stated in Eq. 2. The interplay between the Dirichlet loss and the classification loss is described above. In this section, we give some technical details about the architecture of the network.

**Diffusion pump.** The pump is responsible for generating and updating the matrix of pairwise diffusion distances $\mathcal{D}$. For the generation, we solve any of the following equivalent problems:

$$\min \ \frac{Tr[\mathbf{U}^T \triangle \mathbf{U}^T]}{Tr[\mathbf{U}^T \mathbf{D} \mathbf{U}^T]} \ \equiv \ \max \ \frac{Tr[\mathbf{U}^T \mathbf{A} \mathbf{U}^T]}{Tr[\mathbf{U}^T \mathbf{D} \mathbf{U}^T]} \ , \tag{3}$$

both s.t. $\mathbf{U}^T \mathbf{U} = \mathbf{I}$, where $\mathbf{U}_{n \times p} = f_\theta(\mathbf{A})$, $n = |V|$. Since $\mathbf{D}$ is the diagonal degree matrix, we have $\triangle := \mathbf{D} - \mathbf{A}$. As a result, the min problem *approximates* the $p$ smallest nontrivial eigenvectors of the normalized Laplacian $\tilde{\triangle} := \mathbf{D}^{-1/2} \triangle \mathbf{D}^{-1/2} = \mathbf{I} - \tilde{\mathbf{A}}$, where $\tilde{\mathbf{A}} := \mathbf{D}^{-1/2} \mathbf{A} \mathbf{D}^{-1/2}$ is the normalized adjacency. Equivalently, the max problem *approximates* the $p$ largest nontrivial eigenvectors of the transition matrix $\mathbf{P} = \mathbf{D}^{-1} \mathbf{A}$. Note that $\tilde{\mathbf{A}}$ and $\mathbf{P}$ have the same eigenvectors and also that if $\lambda$ is an eigenvalue of $\mathbf{P}$ then $1 - \lambda$ an eigenvalue for $\tilde{\triangle}$. Note also, that we use "*approximates*" instead of "*finds*". This is due to the limitations of Stochastic Gradient Descent (SGD) when solving the Trace-Ratio problems Wang et al. (2007)Jia et al. (2009)Ngo et al. (2012) in Eq. 3. In this regard, we have the following results with practical implications:

**Theorem 4.1** (Fiedler Environments). *The SGD solution of the Trace-Ratio Min problem in Eq. 3 can be posed in terms of Min $Tr[\mathbf{U}^T (\triangle - \rho \mathbf{D}) \mathbf{U}]$ under orthonormality constraints. This leads to $\triangle \mathbf{U} = \rho^* \mathbf{D} \mathbf{U}$, i.e. to the orthogonal eigenfunctions of the normalized Laplacian $\tilde{\triangle}$ associated with $\rho^*$. However, $\rho^*$ is not necessarily an eigenvalue of $\tilde{\triangle}$, but an approximation of the Fiedler value $\lambda_2$: $\exists \epsilon > 0 : |\lambda_2 - \rho^*| < \epsilon$. As a result, the $p$ columns $\mathbf{u}_i$ of $\mathbf{U}$ satisfy: $\exists \delta > 0 : \|\phi_2 - \mathbf{u}_i\| < \delta$, where $\phi_2$ denotes the Fiedler vector. Then, we obtain what we call a* Fiedler environment.

**Corollary 4.2** (Asymptotic Diffusion Distances). *The norm of $\nabla \mathbf{U}_{ij} := \mathbf{U}_{i:} - \mathbf{U}_{j:}$ (row-difference) is proportional to the approximate commute time between nodes $i$ and $j$, which is $\sum_{t=0}^{\infty} d(i,j)^t$, where $d(i,j) := \|\nabla \mathbf{U}_{ij}\|$. Therefore, the matrix $\mathcal{D}$ relies on Euclidean distances.*

We prove both results in Appendix A. In Appendix B.1, we also give practical evidence of the need to solve Trace-Ratio problems in an SGD context, instead of solving the original Trace problem in Eq. 2. We also justify the convenience of conditioning $\mathbf{U}$ to $\mathbf{A}$, $\mathbf{U} = f_\theta(\mathbf{A})$. Actually, this setting is inspired by how the LINKX method Lim et al. (2021) exploits the graph topology.

**Jumps and Filters.** The bank of learnable structural filters $\{\mathbf{J}^0, \mathbf{J}^1, \ldots, \mathbf{J}^K\}$ is the core of the high-order exploration. Each filter $\mathbf{J}^k = (V, E_k, C_k)$ is an edge-attributed graph whose edges or *support* $E_k = \{(i_k, j_k) \in V \times V : i_k, j_k \in J^k\}$ are given by the pairs of nodes belonging to the jump $J^k$. In Eq. 2, this is implicitly defined with the expression $\mathbf{J}^k = \Pi^k \cdot \exp(-\mathcal{D})$, where $\Pi^k$ is a $\{0,1\}^{n \times n}$ projection matrix defined, for $k > 0$, as follows:

$$\Pi^k(i,j) = \begin{cases} 1 - \Pi^{k-1}(i,j) & \text{if } j \in \text{Idx}[\text{top}k^{-1}(i)] \\ 0 & \text{otherwise} \end{cases} \tag{4}$$

with $\Pi^0(i,j) = \mathbf{I}$ and $\text{top}k^{-1}(i) = \{d(i,j_1), \ldots, d(i,j_k)\}$, where $d(i,j_l) \leq d(i,j_{l+1})$ for $j_l, j_{l+1} \in V$ and $l = 1, 2, \ldots, k-1$. Then, $\text{Idx}[\text{top}k^{-1}(i)]$ are the sorted positions of the distances wrt the node $i$, i.e. the *distance ranks*. In this way, the product $\mathbf{J}^k = \Pi^k \cdot \exp(-\mathcal{D})$ yields $C_k$ and it is derivable wrt $\mathcal{D}$ as in Gao & Ji (2019). Alternatively, we could also rely on the topK network Xie et al. (2020).

**Individual GNNs.** Each structural filter $\mathbf{J}^k$ feeds a vanilla GNN which obtains a partial embedding $\mathbf{H}^k = \sigma(\mathbf{J}^k \mathbf{X} \mathbf{W}^k)$. The GNN also receives the $n \times F$ matrix of node features $\mathbf{X}$ and $\mathbf{x}_i$ denotes the transpose of the $i$−th row of $\mathbf{X}$. Since $\mathbf{J}^k = \Pi^k \cdot \exp(-\mathcal{D})$, then, for a given node $i$, its aggregation is given by $\mathbf{x}_i = \sum_j e^{-d(i,j)} \mathbf{x}_j$ instead of being $\mathbf{x}_i = \mathbf{P}^k \mathbf{x}_i$ as in MixHop Abu-El-Haija et al. (2019) or $\mathbf{x}_i = \left(\sum_k \beta_k \mathbf{P}^k\right) \mathbf{x}_j$ as in Simple Graph Convolution (SGG) Chanpuriya & Musco (2022b).

**Combining GNNs.** Each partial embedding $\mathbf{H}^k = \sigma(\mathbf{J}^k \mathbf{X} \mathbf{W}^k)$ is weighted by a learnable parameter $\alpha_k$, where all the $\alpha_k$ form a convex combination. Then, we concatenate all the weighted embeddings

to form the global embedding $\mathbf{H} := \|_{k=1}^{K} \alpha_k \mathbf{H}^k = \|_{k=1}^{K} \alpha_k \sigma \left( \mathbf{J}^k \mathbf{X} \mathbf{W}^k \right)$. Since $\mathbf{H}$ feeds an MLP in order to minimize the classification loss $\mathcal{L}_c$ as in MixHop, the global embedding tends to retain the best partial embeddings for each node.

**Homophilic Branch.** One limitation of our method is that setting a small value for the hyperparameter $K$ is not enough to deal with homophilic graphs. For this reason, we have added an extra GNN (the homophilic branch) that works as follows: $\mathbf{H}^{HB} = \sigma(\mathbf{A}\mathbf{X}\mathbf{W}^{HB})$. Therefore we concatenate $\mathbf{H} := \mathbf{H}|\alpha_{HB}\mathbf{H}^{HB}$, where $\sum_{k=0}^{K} \alpha_k + \alpha_{HB} = 1$. See, the optimal learned coefficients in Fig. 4 and the complete architecture in Appendix F.

## 5 RELATED WORK

In this paper, we explore High-Order GNNs (HO-GNNs) as a means for addressing heterophily Zhu et al. (2020). One type of HO-GNN results from *rewiring the edges* in the graph. For instance, the method in Bi et al. (2022) explores neighborhoods of several orders (hops) selecting those orders that provide a high correlation between the node features. GATs Veličković et al. (2018) are also a well-known rewiring method: the strength of each edge in the input graph is given by a trainable weight. Diffwire Arnaiz-Rodríguez et al. (2022) is another trainable rewiring method. The basic idea of Diffwire is to estimate the commute-times distance between each pair of nodes and use such a distance matrix to mask the original adjacency matrix. Other non-differentiable rewiring methods are mainly addressed to alleviate the over-squashing issue (bottlenecks obstruct the message-passing process):Topping et al. (2022) and Giovanni et al. (2023).

A second type of HO-GNNs are *Deep/Sequential hop-based* methods, i.e. those models that address the over-smoothing with a deep architecture. GGCNs Yan et al. (2022) attenuate over-smoothing by performing edge correction (corrected edge weights are learned from node degrees, and signed edges are learned from node features). However, Shortest-Paths-MPNNs Abboud et al. (2022) and Ordered-GNNs Song et al. (2023) are more focused on performing robust aggregations. Shortest-Paths-MPNNs compute the shortest paths between any pair of nodes. Then, for each node, several separate aggregations are performed (each one for increasing lengths of the shortest paths); then, the resulting embeddings are weighted. Ordered-GNNs rely on a similar principle: for each node, the hierarchy of a tree rooted in that node is aligned with the hops wrt this node in the graph. As neighboring nodes within $k$ hops form a depth$-k$ subtree, aggregations for shallow sub-trees precede those for deeper ones. Ordered-GNNs introduce a differentiable method for splitting sub-trees.

Finally, *Shallow/Parallel hop-based* methods explore several hop orders in parallel and then integrate the resulting embedding (e.g. via concatenation). MixHop Abu-El-Haija et al. (2019), FSGNNs Maurya et al. (2021) and DualNets Maurya et al. (2022) compute several powers $\mathbf{P}^k$, $k = 1, \ldots, K$ of the normalized adjacency matrix (transition matrix) $\mathbf{P}$. Each power feeds a different GNN. The resulting embeddings are weighed and concatenated for later discrimination. SIGN Frasca et al. (2020) is similar to MixHop but it precomputes the aggregations $\mathbf{P}^k \mathbf{X}$ for the sake of scalability. The Simple Graph Convolution (SGG) method Chanpuriya & Musco (2022b) improves MixHop by learning polynomials of the transition matrix.

Finally, Generalized PageRank GNNs (GPR-GNNs)Chien et al. (2021a)learns jointly the best embedding of each node feature and the best weight of each hop. This is very interesting, since the suitability of $\mathbf{P}^k$, which is encoded by a weight $\gamma_k$, influences the latent space of the features $\mathbf{H}^0 = f_\Theta(\mathbf{X})$ through a learnable function $f_\Theta(.)$. As a result the $k-$th embedding is $\mathbf{H}^k = \mathbf{P}^k \mathbf{H}^0$. This mechanism allows GPR-GNNs to avoid over-smoothing and trade node and topology feature informativeness. However, this strategy produces inconsistent results since GPR-GNNs are better suited for heterophilic graphs, instead of being also useful for homophilic graphs.

**Main Limitation of HO-GNNs.** Most of the existing HO-GNNs explore different powers of the normalized adjacency matrix (transition matrix) $\mathbf{P}$. In other words, they are completely *hop-based*. As a result, the HO-GNNs exploit the labels of the semi-supervised learning process either to alleviate the over-smoothing issue (in the sequential case) or to weigh the importance of each hop order (in the parallel case). However, as the structure of the input graph is *static*, the hops are static as well. Consequently, *the labels cannot be backpropagated to change the structure of the hops, but only the relative importance of each hop or the extent of its aggregation support.*

**Implications.** As a result, dealing with heterophilic graphs goes beyond the potential achievements of hop-based approaches (see our Experiments in section 6). Despite high-order hops being able to connect distant nodes with the same label, such connections can be neither attenuated nor amplified for the sake of classification loss. In this regard, parallel HO-GNNs claim that the powers $\mathbf{P}^k$ can be interpreted as a bank of *structural filters*, i.e. a bunch of aggregators inspired by convolutional filters such as Gabor filters Abu-El-Haija et al. (2019). However, an expressive characterization of a structural filter requires that both its *support* and *coefficients* are learnable.

## 6 EXPERIMENTS AND DISCUSSION

**Experimental settings**.Table 1 presents the results obtained by each model on the standard small-medium datasets Sen et al. (2008) Pei et al. (2020a) Rozemberczki et al. (2019). To ensure consistency, we used the same 10 random splits (48%/32%/20%) provided by Pei et al. (2020a), along with the best configuration for each model. We place the code at https://anonymous.4open.science/r/Diffusion-Jump-GNNs-8EE2/. The above configurations were extracted from Table 1 Di Giovanni et al. (2022) and Tables 3, 10, and 11 in Lim et al. (2021). Overall, our model outperformed all others or achieved a close second place, demonstrating strong competitiveness. We assessed the

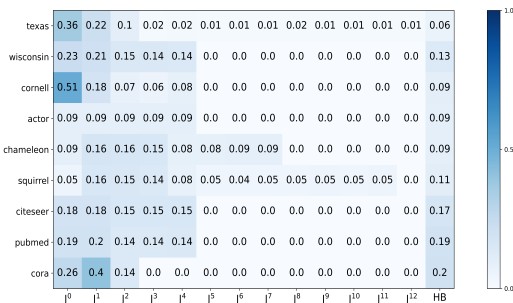

Figure 4: Optimal attention for each jump.

degree of *structural heterophily* using our metric $\mathcal{R}$. For the sake of clarity, we define two HRs: $\mathcal{R} < 8$ indicating mid-low heterophily, and $\mathcal{R} \geq 8$ indicating high heterophily.

Table 1: Node-classification accuracies. Top three models are coloured by **First**, **Second**, **Third**.

| | TEXAS | WISCONSIN | CORNELL | ACTOR | SQUIRREL | CHAMELEON | CITESEER | PUBMED | CORA |
|---|---|---|---|---|---|---|---|---|---|
| HOM LEVEL | 0.11 | 0.21 | 0.30 | 0.22 | 0.22 | 0.23 | 0.74 | 0.80 | 0.81 |
| $\mathcal{R}$ | 18.37 | 6.90 | 6.03 | 209.58 | 20.62 | 8.30 | 5.78 | 7.64 | 7.36 |
| # NODES | 183 | 251 | 183 | 7,600 | 5,201 | 2,277 | 3,327 | 19,717 | 2,708 |
| # EDGES | 295 | 466 | 280 | 26,752 | 198,493 | 31,421 | 4,676 | 44,324 | 5,278 |
| # CLASSES | 5 | 5 | 5 | 5 | 5 | 5 | 5 | 5 | 6 |
| GGCN Yan et al. (2021) | 84.86 ± 4.55 | 86.86 ± 3.29 | 85.68 ± 6.63 | 37.54 ± 1.56 | 55.17 ± 1.58 | 77.14 ± 1.84 | 77.14 ± 1.45 | 89.15 ± 0.37 | 87.95 ± 1.05 |
| GPRGNN Chien et al. (2021b) | 78.38 ± 4.36 | 82.94 ± 4.21 | 80.27 ± 8.11 | 34.63 ± 1.22 | 31.61 ± 1.24 | 46.58 ± 1.71 | 77.13 ± 1.67 | 87.54 ± 0.38 | 87.95 ± 1.18 |
| H2GCN Zhu et al. (2020) | 84.86 ± 7.23 | 87.65 ± 4.89 | 82.70 ± 5.28 | 35.70 ± 1.00 | 36.48 ± 1.86 | 60.11 ± 2.15 | 77.11 ± 1.57 | 89.49 ± 0.38 | 87.87 ± 1.20 |
| GCNII Chen et al. (2020) | 77.57 ± 3.83 | 80.39 ± 3.40 | 77.86 ± 3.79 | 37.44 ± 1.30 | 38.47 ± 1.58 | 63.86 ± 3.04 | 77.33 ± 1.48 | 90.15 ± 0.43 | 88.37 ± 1.25 |
| Geom-GCN Pei et al. (2020a) | 66.76 ± 2.72 | 64.51 ± 3.66 | 60.54 ± 3.67 | 31.59 ± 1.15 | 38.15 ± 0.92 | 60.00 ± 2.81 | 78.02 ± 1.15 | 89.95 ± 0.47 | 85.35 ± 1.57 |
| PairNorm Zhao & Akoglu (2019) | 60.27 ± 4.34 | 48.43 ± 6.14 | 58.92 ± 3.15 | 27.40 ± 1.24 | 50.44 ± 2.04 | 62.74 ± 2.82 | 73.59 ± 1.47 | 87.53 ± 0.44 | 85.79 ± 1.01 |
| GraphSAGE Hamilton et al. (2017) | 82.43 ± 6.14 | 81.18 ± 5.56 | 75.95 ± 5.01 | 34.23 ± 0.99 | 41.61 ± 0.74 | 58.73 ± 1.68 | 76.04 ± 1.30 | 88.45 ± 0.50 | 86.90 ± 1.04 |
| GCN Kipf & Welling (2017) | 55.14 ± 5.16 | 51.76 ± 3.06 | 60.54 ± 5.30 | 27.32 ± 1.10 | 53.43 ± 2.01 | 64.82 ± 2.24 | 76.50 ± 1.36 | 88.42 ± 0.50 | 86.98 ± 1.27 |
| GAT Veličković et al. (2018) | 52.16 ± 6.63 | 49.41 ± 4.09 | 61.89 ± 5.05 | 27.44 ± 0.89 | 40.72 ± 1.55 | 60.26 ± 2.50 | 76.55 ± 1.23 | 87.30 ± 1.10 | 86.33 ± 0.48 |
| MLP | 80.81 ± 4.75 | 85.29 ± 6.40 | 81.89 ± 6.40 | 36.53 ± 0.70 | 28.77 ± 1.56 | 46.21 ± 2.99 | 74.02 ± 1.90 | 75.69 ± 2.00 | 87.16 ± 0.37 |
| CGNN Yamamoto (2019) | 71.35 ± 4.05 | 74.31 ± 7.26 | 66.22 ± 7.69 | 35.95 ± 0.86 | 29.24 ± 1.09 | 46.89 ± 1.66 | 76.91 ± 1.81 | 87.70 ± 0.49 | 87.10 ± 1.35 |
| MixHop Abu-El-Haija et al. (2019) | 77.84 ± 7.73 | 75.88 ± 4.90 | 73.51 ± 6.34 | 32.22 ± 2.34 | 43.80 ± 1.48 | 60.50 ± 2.53 | 76.26 ± 1.33 | 85.31 ± 0.61 | 87.61 ± 0.85 |
| FSGNN(8-hop) Maurya et al. (2021) | 87.30 ± 5.29 | 87.84 ± 3.37 | 85.13 ± 6.07 | 35.75 ± 0.96 | 74.10 ± 1.89 | 78.27 ± 1.28 | 77.40 ± 1.90 | 77.40 ± 1.93 | 87.93 ± 1.00 |
| GRAFF Di Giovanni et al. (2022) | 88.38 ± 4.53 | 88.83 ± 3.29 | 84.05 ± 6.10 | 37.11 ± 1.08 | 58.72 ± 0.84 | 71.08 ± 1.75 | 77.30 ± 1.85 | 90.04 ± 0.41 | 88.01 ± 1.03 |
| LINKX Lim et al. (2021) | 74.60 ± 8.37 | 75.49 ± 5.72 | 77.84 ± 5.81 | 36.10 ± 1.55 | 61.81 ± 1.80 | 68.42 ± 1.38 | 73.19 ± 0.99 | 87.86 ± 0.77 | 84.64 ± 1.13 |
| ACMII-GCN++ Luan et al. (2022) | 88.38 ± 3.43 | 88.43 ± 3.66 | 86.49 ± 6.73 | 37.09 ± 1.32 | 67.40 ± 2.21 | 74.76 ± 2.20 | 77.12 ± 1.58 | 89.71 ± 0.48 | 88.25 ± 0.96 |
| Ordered GNN Song et al. (2023) | 86.22 ± 4.12 | 88.04 ± 3.63 | 87.03 ± 4.73 | 37.99 ± 1.00 | 62.44 ± 1.96 | 72.28 ± 2.29 | 77.31 ± 1.73 | 90.15 ± 0.38 | 88.37 ± 0.75 |
| ASGC Chanpuriya & Musco (2022b) | 85.14 ± 3.06 | 86.06 ± 3.75 | 86.22 ± 3.58 | 36.33 ± 0.79 | 58.38 ± 1.08 | 73.16 ± 1.07 | 66.86 ± 0.86 | 78.72 ± 0.88 | 77.52 ± 1.61 |
| **DJ-GNN** | 92.43 ± 3.15 | 92.54 ± 3.70 | 87.56 ± 1.32 | 36.93 ± 0.84 | 73.48 ± 1.59 | 80.48 ± 1.46 | 77.50 ± 1.33 | 90.08 ± 0.32 | 88.43 ± 0.91 |

**Low Structural Heterophily**. For these datasets, we list their optimal number of jumps (hyperparameter $K$): WISCONSIN ($K = 5$), CORNELL ($K = 5$), CITESEER ($K = 5$), PUBMED ($K = 5$) and CORA ($K = 5$)). We find that a few jumps are enough for achieving or improving the SOTA. However, not all jumps are equally important. For instance, WISCONSIN and CORNELL rely mostly on the first two jumps (Fig. 4), while the remaining datasets rely on the *homophilic branch* (no jump). Actually, CITESEER, PUBMED, and CORA are the datasets with the smallest edge heterophily (HOM LEVEL). In addition, we are the best method in this regime except for CITESEER, where we are very competitive (77.50 (ours) vs 78.02 (GEOM-GCN)). In the only case where we lose, we note that the GEOM-GCN method relies on the geometry of the latent space. In this regard, CITESEER is the dataset with the lowest structural heterophily ($\mathcal{R} = 5.78$), i.e. the geometry of the latent space is a fair representation of the topology of the graph. As a result, adding jumps may complicate that geometry: actually, the most important branches are $\mathbf{J}^0$ and the *homophilic branch* (no jump).

**High Structural Heterophily**. For these datasets Traud et al. (2011) Hu et al. (2020), we also list their optimal number of jumps (hyperparameter $K$): TEXAS ($K = 20$), SQUIRREL ($K = 8$), CHAMELEON ($K = 12$) and ACTOR ($K = 3$). Our best result is for TEXAS ($\mathcal{R} = 18.37$), where we significantly improve the SOTA (92.43 (ours) vs 88.38 (ACMII-GCN++, which is a multi-channel GCN with adaptive channel mixing)). In SQUIRREL, we are slightly outperformed by FSGNN(8-HOP) (73.48 vs 74.10) since our method needs higher frequency eigenfunctions in order to capture the extreme degree variability of this dataset. However, we are very competitive in this dataset since the SQUIRREL graph is very dense and we only need $K = 8$ jumps to achieve good results. We are also the best model in CHAMELEON (whose structural heterophily is the smallest in this set): we obtain 80.48 vs the second-best model FSGNN(8-HOP).

**Parallel (Shallow) vs Sequential (Deep)**. Our method is Parallel (multi-branch shallow GNN) and its performance is the best or it is very competitive in small-medium real-world datasets. There is one exception, the PUBMED dataset, where we obtain 90.09, slightly outperformed by ORDERED GNN with only 5 layers (90.15). This also happens with GCNII which explores 2 to $2^6$ layers. We can conclude that deep methods have a good performance in homophilic datasets but such a performance decays significantly in heterophilic ones.

Table 2: Node-classification accuracies in large graphs. Top three models are coloured by **First**, **Second**, **Third**.

|  | PENN94 | ARXIV-YEAR | OGBN-ARXIV |
|---|---|---|---|
| HOM LEVEL | **0.47** | **0.21** | **0.66** |
| # NODES | 41,554 | 169,343 | 169,343 |
| # EDGES | 1,362,229 | 1,166,243 | 1,166,243 |
| # CLASSES | 5 | 5 | 40 |
| MLP | $73.61 \pm 0.40$ | $36.70 \pm 0.21$ | $55.91 \pm 0.15$ |
| GCN | $82.47 \pm 0.27$ | $46.02 \pm 0.26$ | $59.61 \pm 0.23$ |
| GAT | $81.53 \pm 0.55$ | $46.05 \pm 0.51$ | $60.27 \pm 0.21$ |
| MIXHOP | $83.47 \pm 0.71$ | $51.81 \pm 0.17$ | OOM |
| LINKX | $84.71 \pm 0.52$ | $56.00 \pm 1.34$ | $55.31 \pm 0.81$ |
| **DJ-GNN** | $84.84 \pm 0.34$ | $49.21 \pm 0.20$ | $63.23 \pm 0.12$ |

We have also tested our model in **Very Large Graphs** (see Table 2). In this regard, we note that the memory requirements of our method $- O(n^2)$, where $n$ is the number of nodes $-$, force us to decouple the diffusion pump from the jump exploration. We first learn the matrix of pairwise diffusion distances in an unsupervised way. Later, we use it in a static way to minimize the classification loss. Despite that limitation, we obtain a very competitive performance both for PENN94 (84.84 vs 84.71 with LINKX) and OGBN-ARXIV (63.23 vs 60.27 with GAT). However, our performance decreases in ARXIV-YEAR which is more heterophilic than the others: memory limitations force us to use only $K = 3$ hops for the three datasets.

Finally, we extend our experimental results in Appendixes D (SBMs) and E (hyper-parameters).

## 7 CONCLUSIONS AND FUTURE WORK

In this paper, we propose DIFFUSION-JUMP GNNS, a multi-branch GNN architecture that addresses the heterophily issue from a structural perspective. Firstly, we define node-classification in terms of a Dirichlet problem. This allows us to define a new measure of heterophily: structural heterophily. Having this measure in mind we formulate a loss function that governs the interplay between the two main components of our architecture: the **diffusion pump** (which generates diffusion distances) and the **parallel jumps** (which drive the exploration of links between nodes with similar labels). The most important contribution of our model is that the diffusion distances, and consequently the jumps and the structural filters derived from them, are fully learnable. Our experiments show that our model outperforms the SOTA or it is very competitive. Finally, our future work includes: a) scalability, in terms of memory, b) automatic jump selection, and c) improving SGD for Trace-Ratio problems.

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

## A  Inspiring Methods

We devote this appendix to introduce some **links with very inspiring methods** in the literature. For instance, our Dirichlet formulation is inspired by classical graph-based semi-supervised methods. In particular, the work in Zhou et al. (2003) poses the problem of propagating known labels $\ell(B)$ to unknown nodes $u \in U$. Let $\mathbf{Y}$ be a $n \times c$ matrix where $\mathbf{Y}(i,j) = 1$ means that node $i \in B$ has label $j$ and $\mathbf{Y}(i,j) = 0$ otherwise. Then, we have the following result:

**Theorem A.1** (Dichilet label propagation Zhou et al. (2003)). *The optimal label of each node $i$ is given by $\mathbf{y}(i) = \arg\max_{j \leq c} \mathbf{F}(i,j)$, where $\mathbf{F} = \beta\left(\mathbf{I} - \alpha\mathbf{P}\right)^{-1}$, being $\mathbf{P}$ the transition matrix and $\alpha + \beta = 1$. In addition, $\mathbf{F}$ minimizes $\frac{1}{2}\left(Tr[\mathbf{F}^T\triangle\mathbf{F}] + \mu\sum_i\|\mathbf{F}_{i:} - \mathbf{Y}_{i:}\|^2\right)$, where $\mu > 0$ is a regularization parameter satisfying $\alpha = 1/(1 + \mu)$.*

Consequently, the diffusion pump in our model is governed by a similar equation: Eq. 2. We prove the above theorem and its relationship with absorbing random walks Doyle & Snell (2000) and semi-supervised image segmentation Grady (2006) in Appendix C.

Finally, another important source of inspiration was the design of escape probabilities in terms of diffusion equations. Actually, there is a substantial body of theory linking spectral clustering, random walks, diffusion distances, and meta-stable states Meilă & Shi (2001)Nadler et al. (2005)Nadler & Galun (2006) to be analyzed also in the same appendix. Herein, we only highlight the following result:

**Theorem A.2** ( Nadler & Galun (2006)). *Given a probability function in Boltzmann form $p(\mathbf{x}) = e^{-U(\mathbf{x})}$ in a given latent space $\mathbf{X}$, the random walk with transition matrix $\mathbf{P}$ converges to the stochastic differential equation $\dot{\mathbf{x}}(t) = -\nabla U(\mathbf{x}) + \sqrt{2}\dot{\mathbf{w}}(t)$, where $\mathbf{w}$ denotes Brownian motion. Also, the potential time scales describing the expected time of passage between clusters rely on the potential function $U(\mathbf{x})$.*

We expand this result, and more links with meta-stable states, in Appendix B.4.

## B  Formal Results with Practical Implications

For the sake of clarity, in this appendix we develop the key concepts of the theorems stated in the paper instead of providing detailed proofs. Our emphasis here is on the practical implications of each result. For more details, we refer the reader to the cited papers.

### B.1  Trace Ratio and SGD

**The Trace Ratio Problem.** Min $\mathbf{U}^T\triangle\mathbf{U}$ is achieved by $\mathbf{V}_{n \times p}$ whose $p$ columns are given by the eigenvectors of the $p$ smallest eigenvalues $\lambda_1 \leq \lambda_2 \leq \ldots \leq \lambda_p$ of the Laplacian $\triangle$. Then, $\text{Tr}[\mathbf{V}^T\triangle\mathbf{V}] = \lambda_2 + \ldots + \lambda_p$, when the graph $G = (V, E)$ with adjacency matrix $\mathbf{A}$ is connected. However, $\mathbf{V}$ does not necessarily minimize $\text{Tr}[\mathbf{V}^T\mathbf{D}\mathbf{V}]$. As a result, we have that

$$\rho^* := \text{Min}_{\mathbf{U}^T\mathbf{U}=\mathbf{I}} \frac{\text{Tr}[\mathbf{U}^T\triangle\mathbf{U}]}{\text{Tr}[\mathbf{U}^T\mathbf{D}\mathbf{U}]} \leq \frac{\text{Tr}[\mathbf{V}^T\triangle\mathbf{V}]}{\text{Tr}[\mathbf{V}^T\mathbf{D}\mathbf{V}]} \leq \frac{\lambda_2 + \ldots + \lambda_p}{d_1 + \ldots + d_p} , \tag{5}$$

where $d_1 \leq d_2 \leq \ldots \leq d_n$ are the sorted degrees. As a result, we have the following bounds:

$$\frac{\lambda_2 + \ldots + \lambda_p}{d_{p+1} + \ldots + d_n} \leq \rho^* \leq \frac{\lambda_2 + \ldots + \lambda_p}{d_1 + \ldots + d_p} . \tag{6}$$

The definition of $\rho^*$ plays a key role in the original trace-ratio optimization. Following Ngo et al. (2012), such a problem is formulated in scalar terms, i.e. in terms of finding

$$\rho^* = \arg\min_{\mathbf{U}^T\mathbf{U}=\mathbf{I}} f(\rho) := \text{Tr}[\mathbf{U}^T\triangle\mathbf{U}] - \rho\text{Tr}[\mathbf{U}^T\mathbf{D}\mathbf{U}] . \tag{7}$$

Actually, for $\rho^*$ we have have that

$$\text{Min}_{\mathbf{U}^T\mathbf{U}=\mathbf{I}}\text{Tr}[\mathbf{U}^T(\triangle - \rho^*\mathbf{D})\mathbf{U}] = 0 . \tag{8}$$

Therefore, the trace-ratio problem can be solved by alternating two updating steps:

$\mathbf{U}$ : Given $\rho$, apply the Lanczos method to obtain the $p$ largest eigenvalues of the transition matrix $\mathbf{P} - \rho\mathbf{D}$ (the smallest of $\triangle - \rho\mathbf{D}$) and their associated eigenvectors $\mathbf{U}$.

$\rho$ : Given the current eigenvectors $\mathbf{U}$, update $\rho = \frac{\text{Tr}[\mathbf{U}^T \triangle \mathbf{U}]}{\text{Tr}[\mathbf{U}^T \mathbf{D} \mathbf{U}]}$

In the above process, the update of $\mathbf{U}$ ensures the orthogonality constraint.

**The Trace Ratio and SGD.** However, solving the trace-ratio problem through gradient descent drives us to a different solution from the eigenvectors of $\triangle - \rho^*\mathbf{D}$. For instance, consider the Dirichlet loss $\mathcal{L}_D = \frac{\text{Tr}[\mathbf{U}^T \triangle \mathbf{U}]}{\text{Tr}[\mathbf{U}^T \mathbf{D} \mathbf{U}]}$. Then, its gradient (supposing that the orthogonality is enforced by a complementary loss) is given by:

$$\nabla\mathcal{L}_D := \frac{2\triangle\mathbf{U} - 2\rho\mathbf{D}\mathbf{U}}{\text{Tr}[\mathbf{U}^T\mathbf{D}\mathbf{U}]} \ . \tag{9}$$

Therefore, $\nabla\mathcal{L}_D = 0$ implies $\triangle\mathbf{U} = \rho^*\mathbf{D}\mathbf{U}$, where $\rho^* \to 0$ is the asymptotic value of the trace ratio. As a result, we have that the optimal $\mathbf{U}$ satisfy $\tilde{\triangle}\mathbf{U} = \rho^*\mathbf{U}$, i.e. the gradient descent converges to the (orthonormal) functions of the normalized Laplacian $\tilde{\triangle}$ associated with the value $\rho^*$. However, as $\rho^*$ is not necessarily an eigenvalue of $\tilde{\triangle}$, but it is close to the Fiedler value $\lambda_2$, we denote $\mathbf{U}$ as a **Fiedler environment**. It is an environment since the $p$ columns $\mathbf{u}_{:j}$ are mutually orthonormal and close to the Fiedler vector $\phi_2$ insofar their Dirichlet energies $\rho_j^* = \mathbf{u}_{:j}^T\tilde{\triangle}\mathbf{u}_{:j}$ satisfy $|\lambda_2 - \rho_j^*| < \epsilon$ with $\epsilon \to 0$ (**Theorem** 4.1).

In our experiments, we have chosen the trace-ratio formulation because:

a) It leads an **implicit normalization** of the gradient $\nabla\mathcal{L}_D$, namely $\text{Tr}[\mathbf{U}^T\mathbf{D}\mathbf{U}]$.

b) The **gradient is more structured** when we apply the constrain $\mathbf{U} = f_\theta(\mathbf{A})$, where $\mathbf{A}$ is the adjacency matrix.

Regarding **a)**, our implicit normalization alleviates the problem of landing in local minima due to the orthonormalization constraint (that we also enforce in the global loss). As noted Edelman et al. (1998), constraints of the form $\mathbf{U}^T\mathbf{U} = \mathbf{I}$ define a Riemannian manifold and the trace problem s.t. them is not geodesically convex. In Xu et al. (2018) this is addressed by introducing a Riemannian gradient and retraction normalization.

However, our main gain in performance is achieved when we address **b)** via the joint effect of normalization and $\mathbf{U} = f_\theta(\mathbf{A})$. In our preliminary experiments, we compared the gradient when applying the constraint $\mathbf{U} = f_\theta(\mathbf{A})$ vs the one when doing only $\mathbf{U} = f_\theta(\mathbf{I})$. Discarding the biases, and the non-linearities in both cases, we have $\mathbf{U} = \mathbf{A}\mathbf{W}$ vs $\mathbf{U} = \mathbf{W}$. For simplicity, we consider the gradient wrt a single column, i.e. we analyze $\mathbf{u} = \mathbf{A}\mathbf{w}$ vs $\mathbf{u} = \mathbf{w}$

$$\nabla\mathcal{L}'_{D,\theta} := \frac{2(\triangle - \rho\mathbf{D})(\mathbf{A}\mathbf{w})}{\text{Tr}[(\mathbf{A}\mathbf{w})^T\mathbf{D}(\mathbf{A}\mathbf{w})]} \quad \text{vs} \quad \nabla\mathcal{L}_{D,\theta} := \frac{2(\triangle - \rho\mathbf{D})\mathbf{w}}{\text{Tr}[\mathbf{w}^T\mathbf{D}\mathbf{w}]} \ . \tag{10}$$

Given a random initialization of $\mathbf{w}$, this vector plays the role of a random projector of the rows in $\mathbf{A}$. Following, the Johnson-Lindenstrauss Lemma, $\hat{\mathbf{w}} = \mathbf{A}\mathbf{w}$ tends to replicate the structure of the adjacency. Actually, if the entries of $\mathbf{w}_i \sim \mathcal{N}(0,1)$ then, those of the projection satisfy $\hat{\mathbf{w}}_i \sim \mathcal{N}(0, d_i^2)$, where $d_i$ is the degree of node $i$. As a result, if we have $c$ well-defined communities in the graph $G = (V, E)$ with adjacency matrix $\mathbf{A}\mathbf{w}$, then the projection $\hat{\mathbf{w}}$ is near piecewise constant (actually the norm of the $i-$th row is preserved: $\|\mathbf{A}_{i:}\| \approx \|\hat{\mathbf{x}}\|$). As a result, the projection $\hat{\mathbf{w}}$ is more structured than $\mathbf{w}$ and this is propagated and even amplified during gradient descent. In addition, the normalization of $\nabla\mathcal{L}'_{D,\theta}$ is stronger than that of $\nabla\mathcal{L}_{D,\theta}$.

Overall, when evaluating the performance in SQUIRREL and CHAMELEON using only $\mathbf{U} = f_\theta(\mathbf{I})$ (i.e. using $\nabla\mathcal{L}_{D,\theta}$) we only obtain $41.38 \pm 2.98$ and $58.48 \pm 4.69$. However, using $\mathbf{U} = f_\theta(\mathbf{A})$ (gradient $\nabla\mathcal{L}'_{D,\theta}$) leads to $73.48 \pm 1.59$ and $80.48 \pm 1.46$ respectively.

Finally, a detailed impact of the two above formulations in the variance of the SGD as in An et al. (2021) is beyond the scope of this paper.

## B.2 DIFFUSION DISTANCES

As explained above, optimizing the Dirichlet loss leads to Fiedler environments, i.e. the rows $\mathbf{U}_{i:}$ contain the $p$ nearly orthogonal eigenvectors with eigenvalues $\gamma_1 = 1 > \gamma_2 \geq \ldots \geq \gamma_p$. Following Nadler et al. (2005), the *diffusion distance* $D_t(i, j)$ at time $t$ between two nodes $i$ and $j$ is defined spectrally as:

$$D_t^2(i,j) := \sum_k \frac{1}{\pi(k)} \left(p(k,t|i) - p(k,t|j)\right)^2 = \Gamma^{2t} \|\mathbf{U}_{i:}^* - \mathbf{U}_{j:}^*\|^2 , \qquad (11)$$

where $\pi(k) := d_k/vol(G)$ are the components from the stationary probability distribution $\lim_{t \to \infty} p(j, t|i) = \pi$, i.e. the eigenvector $\mathbf{U}_{:1}^*$ corresponding to $\gamma_1 = 1$. In the above equation, $\mathbf{U}^*$ denote the true eigenvectors of the transition matrix $\mathbf{P}$, and $\Gamma := \text{diag}(\gamma_1, \gamma_2, \ldots, \gamma_p)$ is the diagonal matrix with the corresponding eigenvalues. Thus, Eq. 11 can be explained in the following terms:

**a)** $D_t^2(i, j)$ compare the probabilities that two random walks (one starting in $i$ and another one in $j$) reach any other node $k$ in time $t$.

**b)** The spectral interpretation relies on the spectral theorem applied to the transition matrix $\mathbf{P} = \mathbf{U}^* \Lambda \mathbf{U}^{*T}$. As a result, $\mathbf{P}^t = \mathbf{U}^* \Gamma^t \mathbf{U}^{*T} = \sum_{r=1}^n \gamma_r^t \mathbf{U}_{:r}^* \mathbf{U}_{:r}^{*\,T}$, with $n = |V|$.

However, since determining what is the correct diffusion time is very hard (it is usually a hyper-parameter in some GNNs), we are interested in the asymptotic diffusion distance $D_{t \to \infty}^2$. Qiu and Hancock Qiu & Hancock (2007) determined that

$$\sum_{t=0}^\infty D_t^2(i,j) = \sum_{r=2}^n \frac{1}{1 - \gamma_r} \left(\mathbf{U}_{ri}^* - \mathbf{U}_{rj}^*\right)^2 \qquad (12)$$

i.e. eigenvalues $\{1 - \gamma_r\}$ of the normalized Laplacian $\tilde{\triangle}$ are used instead of those of $\mathbf{P}$. Actually, the right side of the above equation is the well-known *commute times* Chandra et al. (1989) distance $\text{CT}(i, j)$. Note that such a distance is dominated by the Fiedler value and vector: $\lambda_2 = (1 - \gamma_2)$ and $\mathbf{U}_{:2}^*$, respectively. This fact simplifies the interpretation of our approximate diffusion distance as follows:

**a)** Our approximated eigenvectors, contained in the $p$ columns of $\mathbf{U}$ have eigenvalues (Dirichlet energies) $\rho_r^*$ close to $\rho^*$ (the optimal trace ratio achieved by the Dirichlet loss).

**b)** Theorethically, we have that the smallest $\rho_{r_{\min}}^*$ satisfies $\lambda_2 \leq \rho_{r_{\min}}^*$. Therefore, if we order $\rho_r^*$ in ascending order, then we obtain

$$\sum_{t=0}^\infty D_t^2(i,j) \approx \sum_{r=1}^p \frac{1}{\rho_r^*} \left(\mathbf{U}_{ri} - \mathbf{U}_{rj}\right)^2 \approx \text{CT}(i,j) . \qquad (13)$$

**c)** However, in the heterophilic regime (where the labels break the structure) we usually have $\lambda_2 \ll \rho_{r_{\min}}^*$. See for instance the Fiedler environments obtained for SBMs in Figure 5 and the discussion below (Appendix D). As a result, in practice we have

$$d(i,j) := \sum_{r=1}^p \left(\mathbf{U}_{ri} - \mathbf{U}_{rj}\right)^2 = \|\mathbf{U}_{i:} - \mathbf{U}_{j:}\|^2 = \alpha \text{CT}(i,j) \text{ where } \alpha \ll 1. \qquad (14)$$

This proofs **Corollary** 4.2.

## B.3 ESCAPE PROBABILITIES

Approximating commute times distances is very convenient for our jump-based analysis, since it is well known that the *escape probability* is related to the commute times distance Doyle & Snell (2000): $p_{esc} = \frac{1}{\text{CT}(i,j)}$. Escape probabilities are actually dependent on the spectral gap (approximated by the Fiedler value $\lambda_2$). This is illustrated in the very first Figure of this paper (Figure 2), where a random walker tries to escape from the community $\bar{A}$. A classic result Meilă & Shi (2001) shows that

the probability that a random walk started in its asymptotic ($t \to \infty$) distribution $\pi$ is transitioning from $i \in \bar{A}$ to $j \in A$ in one step is $p_{esc}(\bar{A}, A) = \frac{\text{cut}(\bar{A}, A)}{\text{vol}(\bar{A})}$, where $\text{cut}(\bar{A}, A) = \sum_{i \in \bar{A}, j \in A} e_{ij}$ (in the Figure we have that $\text{cut}(\bar{A}, A) = 1$).

Therefore, as $d(i, j) = \alpha \text{CT}(i, j)$, with $\alpha \ll 1$ our jump-hierarchy is closer to that of the escape probability than choosing $\text{CT}(i, j)$ as asymptotic diffusion distance. In addition, we are sensitive to the spectral gap since the Fiedler environment contains approximations of the Fiedler vector, and the spectral gap is approximated in turn by the Dirichlet energy of the Fiedler vector.

### B.4 CLUSTERING AND METASTABLE STATES

Minimizing the Dirichlet loss in conjunction with the classification loss (see Eq 2) leads to a trade-off between two *clustering* problems. On the one hand, we infer a piecewise-smooth latent space. On the other hand, we simultaneously try to preserve the structure of the input graph as much as possible. In both cases, we try to find metastable states. A metastable state is a concept borrowed from dynamical systems but basically, it is an equilibrium state in a random process (for instance the one defined by a random walk that tries to escape from a community in Figure 2). Metastable states are also characterized by wells in potential functions $U(\mathbf{x})$, where $\mathbf{x}$ is a state and its probability is given by the Boltzmann distribution $p(\mathbf{x}) = e^{-U(\mathbf{x})}$. Then, the characteristic relaxation processes and time scales of a given space are usually described by a Stochastic Diffusion Equation (SDE):

$$\dot{\mathbf{x}}(t) = -\nabla U(\mathbf{x}) + \sqrt{2}\dot{\mathbf{w}}(t) \tag{15}$$

where $\mathbf{w}$ denotes Brownian motion. In the above equation, we have a *drag* term (the gradient) that drives the process to a deep well, and a *random* term (the Brownian motion) that allows us to escape from local minima. During this process, we find different time scales: fast scales while we are moving through a given well, and slow scales when we try to escape from it. For instance, escaping from the right community in Fig. 2 takes a long time which depends on the difference between the potential at the well $U(\mathbf{x}_{min})$ and that of the saddle point $U(\mathbf{x}_{max})$ Nadler & Galun (2006). This time is in turn the inverse of the spectral gap, i.e. there is a spectral interpretation of the SDE. Such interpretation comes from the analysis of the Fokker-Planck equation:

$$\partial_t p(\mathbf{x}, t) = \nabla \cdot [\nabla p(\mathbf{x}, t) + p(\mathbf{x}, t)\nabla U(\mathbf{x})] \ , \tag{16}$$

This equation leads to the pdf of the SDE and it has a spectral interpretation. More precisely, the eigenvectors of $\mathbf{P}$ converge to the eigenfunctions $\Psi(\mathbf{x})$ of the Fokker-Planck equation as follows Nadler & Galun (2006)Nadler et al. (2006):

$$\tilde{\nabla}\Psi(\mathbf{x}) := \triangle\Psi - \nabla\Psi \cdot \nabla U = -\mu\Psi(\mathbf{x}) \ , \tag{17}$$

where $\triangle = \nabla \cdot \nabla$ is the Laplacian and $\mu$ the eigenstates (eigenvalues). As a result, we may use the Fiedler vector to characterize the separation between two clusters. The steepest the Fiedler vector, the better the separation (**Theorem** A.2). Interestingly, the third eigenvector $\tilde{\nabla}$ may not work well as a state separator when we have different spatial scales Nadler & Galun (2006).

## C DIRICHLET LABEL PROPAGATION

Herein, we expand the Dirichlet methods for semi-supervised learning cited in Appendix A. In particular, the work in Zhou et al. (2003) poses the problem of propagating known labels $\ell(B)$ to unknown nodes $u \in U$. Let $\mathbf{Y}$ be a $n \times c$ matrix where $\mathbf{Y}(i, j) = 1$ means that node $i \in B$ has label $j$ and $\mathbf{Y}(i, j) = 0$ otherwise. Then, the optimal label of each node $i$ is given by $\mathbf{y}(i) = \arg\max_{j \leq c} \mathbf{F}(i, j)$, where

$$\mathbf{F} = \beta \left(\mathbf{I} - \alpha\mathbf{P}\right)^{-1} \ , \tag{18}$$

being $\mathbf{P}$ the transition matrix and $\alpha + \beta = 1$. The $n \times c$ matrix $\mathbf{F}$ works as a basic node representation (not exactly a latent space) since each of is $c$ rows is stochastic. However, its construction exploits the powers of $\mathbf{P}$ as follows:

$$\mathbf{F}(t) = (\alpha\mathbf{P})^{t-1}\mathbf{Y} + (1 - \alpha)\sum_{i=0}^{t-1}(\alpha\mathbf{P})^i\mathbf{Y} \ . \tag{19}$$

and

$$\mathbf{F} = \lim_{t \to \infty} \mathbf{F}(t) = (1 - \alpha)(\mathbf{I} - \alpha\mathbf{P})^{-1} \ . \tag{20}$$

In addition, we also noted that $\mathbf{F}$ is also the solution of the Dirichlet problem:

$$\text{Min} \ \ \mathcal{L}_Q = \frac{1}{2} \left( \text{Tr}[\mathbf{F}^T \triangle \mathbf{F}] + \mu \sum_i \|\mathbf{F}_{i:} - \mathbf{Y}_{i:}\|^2 \right) \ . \tag{21}$$

where $\mu > 0$ is a regularization parameter satisfying $\alpha = 1/(1 + \mu)$. The proof is obtained by setting the gradient to zero:

$$
\begin{aligned}
\nabla \mathcal{L}_{Q_{\mathbf{F}}} &= \mathbf{F} - \mathbf{PF} + \mu(\mathbf{F} - \mathbf{Y}) \\
&= \mathbf{F} - \frac{1}{1 + \mu}\mathbf{PF} - \frac{\mu}{1 + \mu}\mathbf{Y} \\
&= \mathbf{F} - \alpha\mathbf{PF} - \beta\mathbf{Y} = 0 \ ,
\end{aligned}
\tag{22}
$$

which leads to Eq. 19.

Concerning the relationship of this formulation with absorbing random walks Doyle & Snell (2000), the main idea is to extend the $n \times n$ transition matrix $\mathbf{P}$ so that:

**a)** We include an upper block with the $p \times p$ identity matrix $\mathbf{I}$. This block represents the $p$ absorbing states, where $p = |\ell(B)|$. Then the $n \times p$ block $\mathbf{R}$ encodes the prior probabilities of reaching an absorbing state from a non-absorbing one.

**b)** The absorbing probabilities are given by $\mathbf{B} := (\mathbf{I}_{n \times n} - \mathbf{P})^{-1}\mathbf{R}$ .

Finally, the random-walker version Grady (2006) is quite similar to the above one, but reorganizes the Laplacian matrix (**Theorem** A.1).

## D   SBM ANALYSIS

The following experiment aims to illustrate the interplay between our novel measure of structural heterophily $\mathcal{R}$ and the extent of the spectral gap. We also show the Fiedler Environments and how they are influenced by the classification loss (labels). For each heterophilic regime, we show both the corresponding pairwise distance matrix (diffusion map) and the resulting homophiliation. We have depicted in Figure 5 the main ingredients of our approach as a means of illustrating some technical details introduced in *Appendix A*. In particular, when analyzing SBM graphs under structural heterophily we observe several interesting phenomena.

**Original vs Learned**. Instead of precalculating the eigenvectors, as in Directional GNNs Beaini et al. (2021), we learn them. Our learned (approximate) eigenvectors are relatively close to the Fiedler vector (in terms of how they discriminate the two classes). This is what we call *Fiedler Environments* but, in a semi-supervised setting, i.e. the learned eigenvectors are reactive both to the Dirichlet loss and to the classification loss. Despite being noisy, the vectors in the Fiedler Environments are able of partitioning a class when needed (especially for high values of $\mathcal{R}$).

**Diffusion Map.** Our pairwise distances are also reactive to semi-supervised classification. However, the Dirichlet loss tends to flatten the intra-communities distances as much as possible. Flattening is a mechanism to enforce intra-community diffusion in the homophilic regime. In the heterophilic regime, however, the diffusion map enforces exploration via high-order jumping (see lateral steps in the blue region and the loss of the red peak in the small community).

**Embedding.** We can also see how the embedding is affected by structural heterophily. When we have a structural cluster with nodes of two classes, the respective embeddings are correctly separated, but the margin of this separation decreases as $\mathcal{R}$ increases. This can be seen in graphs that have high $\mathcal{R} \gg 1$, where it is common to find subclusters of nodes that belong to the other classes.

**Interplay between $\mathcal{R}$ and the gap.** Finally, we extend the experiments of Chanpuriya & Musco (2022a) by incorporating a third axis in addition to variate $p$ and $q$. This new axis is the structural heterophily. We proceed as follows. We generate four *basic SBMs* attending to increasing spectral gaps: $\frac{p-q}{p+q} \in \{0.2, 0.5, 0.67, 0.98\}$. For each basic SBM we have generated six levels of increasing

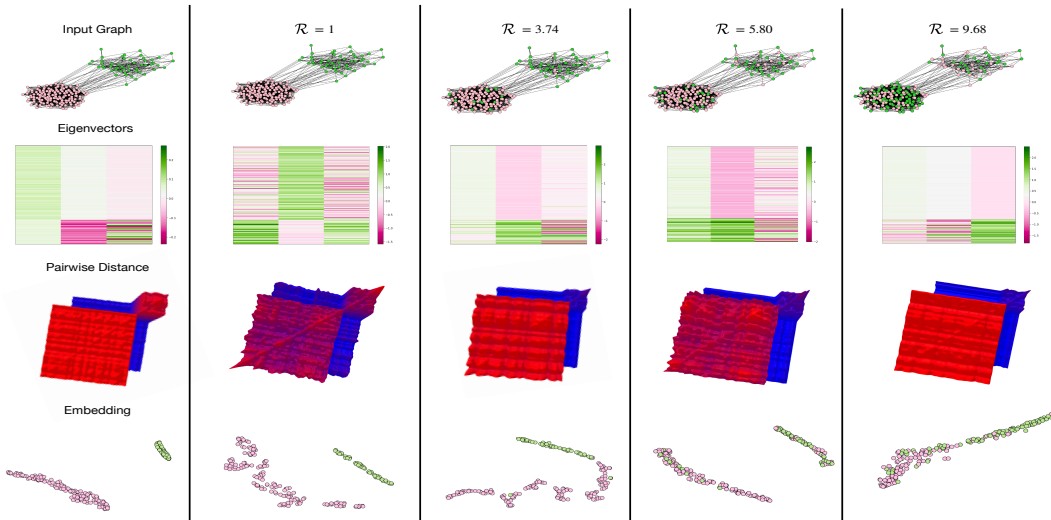

Figure 5: Structural Heterophily in SBMs. Left: The original homophilic graph (First row), its $p = 3$ eigenvectors (Second row), the pairwise distance matrix (Third row), and the resulting embedding. The remaining columns to the right have the same structure for increasing levels of structural heterophily. Note the evolution of the Fiedler Environments and the homophiliations. In all cases, we use $K = 10$ jumps.

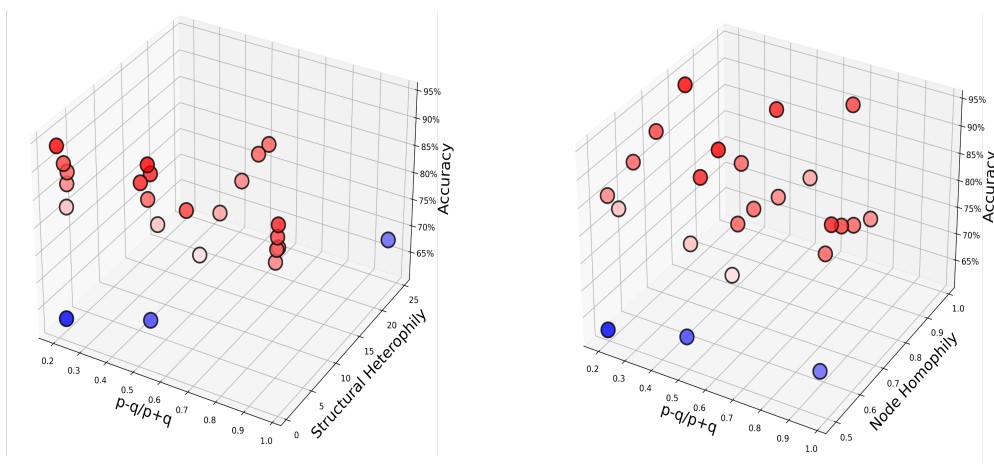

Figure 6: Interplay between heterophily and the spectral gap $\frac{p-q}{p+q}$. Left: Results wrt structural heterophily. Right: Results wrt node homophily.

structural heterophily $\mathcal{R}$. In parallel, we also generate six levels of increasing node homophily as a means of complementing structural heterophily.

We show our results in Figure 6.

a) **Small Gaps help.** Our method is based on spectral clustering, which means that keeping the gap low factor is key. This helps our method to choose whether to jump outside the cluster

> looking for a node with the same label (Heterophilic regime) or to stay and only look around (homophilic regime). This common case is supported by our method without problems.

    **b) Low/Medium Structural Heterophily.** If the structure is quite correlated with the label and the spectral gap is not too high, our method is able of achieving good results even when the structure is noisy.

    **c) Large Gaps lead to oversmoohing.** Our worst performance is achieved when the inter-class message passing is massive. This leads to oversmothing due to the high connectivity of the graph. This high connectivity cannot be controlled by our pump (see the blue dots).

We have also performed the same experiment, but changing the structural heterophily measure to node homophily, in order to display the difference between both. Note that our measure fails when the spectral gap is large. This happens because the Dirichlet energy in a near-complete graph is minimal. This lack of structure leads $\mathcal{R}$ to consider that all the nodes are in the same cluster, i.e. is no heterophily).

# E  EXPERIMENTAL AND COMPUTATIONAL DETAILS

In this section, we provide details about the datasets (see Table 3) and all the parameters and configurations of our experiments (see Table 4 in order to clarify the architecture and the results better. DIFFUSION-JUMP GNNs is implemented in PyTorch Paszke et al. (2019), using PyTorch Geometric Fey & Lenssen (2019) and ogbn datasets Hu et al. (2020). For reproducibility, code, and instructions are available on our GitHub with all the selected configurations and logs. We have also included the computational (See Figure 7) in order to clarify the derivability of $topk$ in PyTorch.

Table 3: Statistics of the datasets used in our experiments.

| DATASET | AVG D | DENSITY | NODE H | CLASS H |
|---|---|---|---|---|
| TEXAS | 1.77 | 0.0090 | 0.07 | 0.001 |
| WISCONSIN | 2.05 | 0.0080 | 0.17 | 0.094 |
| CORNELL | 1.62 | 0.0080 | 0.11 | 0.047 |
| ACTOR | 3.94 | 0.0005 | 0.16 | 0.011 |
| SQUIRREL | 41.73 | 0.0080 | 0.09 | 0.025 |
| CHAMELEON | 15.85 | 0.0070 | 0.10 | 0.062 |
| CITESEER | 2.73 | 0.0008 | 0.71 | 0.627 |
| PUBMED | 4.49 | 0.0002 | 0.79 | 0.664 |
| CORA | 3.89 | 0.0014 | 0.83 | 0.776 |
| PENN94 | 3.89 | 0.0014 | 0.83 | 0.776 |
| OGBN-ARXIV | 7 | 0.0004 | 0.66 | 0.444 |
| ARXIV-YEAR | 7 | 0.0004 | 0.22 | 0.272 |

In the following Table 4, we include the hyperparameters that have yielded the best results during the experimentation phase. It is worth noting that the experiments were conducted using the same 10 random splits as in Pei et al. (2020c), training during 700 epochs and utilizing early stopping.

Table 4: Best hyperparameters for our datasets.

| DATASET | HIDDEN CHANNELS | DROPOUT | LR | WEIGHT DECAY | K/#JUMPS |
|---|---|---|---|---|---|
| TEXAS | 64 | 0.2 | 0.03 | 0.0005 | 20 |
| WISCONSIN | 64 | 0.5 | 0.03 | 0.0005 | 5 |
| CORNELL | 128 | 0.5 | 0.03 | 0.001 | 5 |
| ACTOR | 16 | 0.2 | 0.03 | 0.0001 | 3 |
| SQUIRREL | 128 | 0.5 | 0.003 | 0.0005 | 8 |
| CHAMELEON | 128 | 0.35 | 0.003 | 0.0005 | 12 |
| CITESEER | 128 | 0.5 | 0.003 | 0.0005 | 5 |
| PUBMED | 128 | 0.3 | 0.01 | 0.0005 | 3 |
| CORA | 128 | 0.5 | 0.002 | 0.0005 | 5 |
| PENN94 | 16 | 0.5 | 0.001 | 0.0001 | 3 |
| OGBN-ARXIV | 128 | 0.3 | 0.01 | 0.0005 | 3 |
| ARXIV-YEAR | 128 | 0.2 | 0.003 | 0.0005 | 3 |

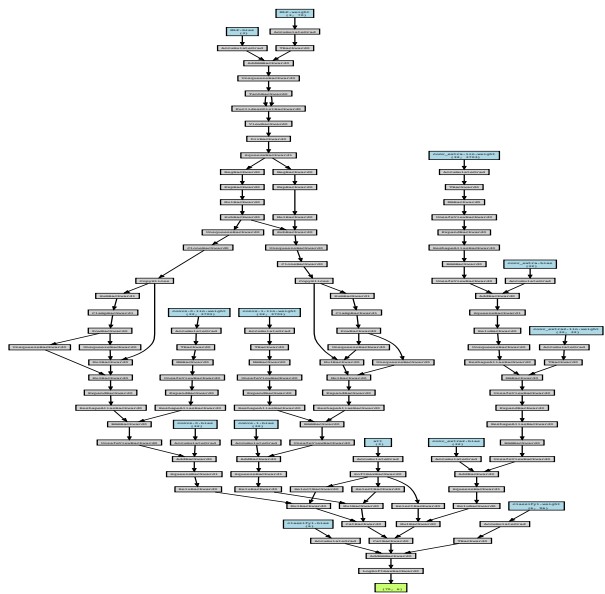

Figure 7: The Computational Graph for $K = 3$ jumps. All branches depend on the diffusion pump (top-left) except $HB$ (the Homophily Branch, top-right).

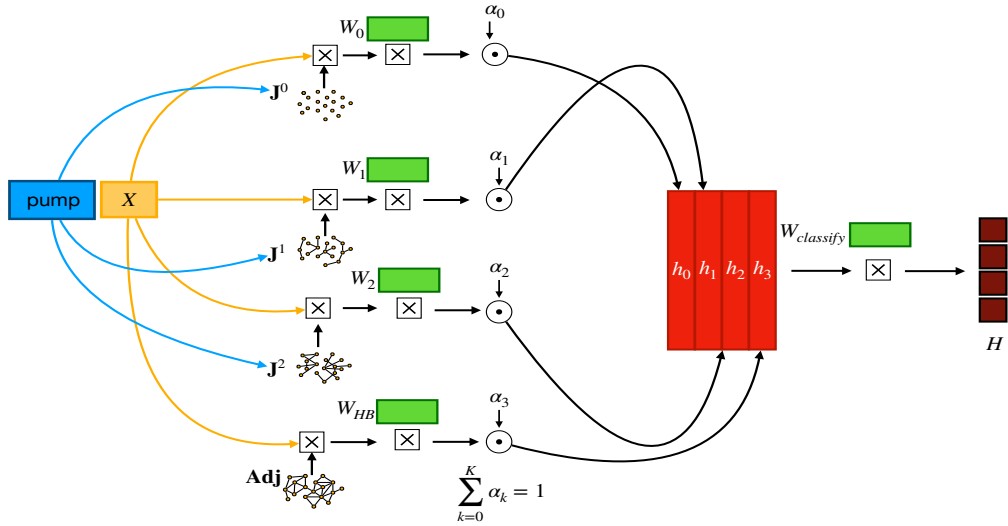

Figure 8: Proposed architecture, for $k = 3$, where we can see 3 branches that use the structural filters $\{\mathbf{J}^0, \mathbf{J}^1, \mathbf{J}^2\}$, and one additional branch that uses the initial adjacency (the homophilic branch).

## F  THE ARCHITECTURE OF DIFFUSION-JUMP GNNs

DIFFUSION-JUMP GNNs have the following elements (see Fig. 8):

**a)** We have a **diffusion pump** (in blue) which generates *diffusion distances* $d(i, j)$ by learning the nontrivial top eigenvectors of $\mathbf{P}$ subject to the labeling of the training set. These distances are adjusted during backpropagation.

**a)** Given the diffusion distances $d(i, j)$ we compute the *jump hierarchy* (see Figure 2). Given a node $i$ we have that $i \in J_i^0$ ($k = 0$) is the closest node wrt itself, $j_1 \in J_i^1$ ($k = 1$) are nodes so that only $j_1$ is closer to node $i$ than any of them, $j_2 \in J_i^2$ ($k = 2$) are nodes so that only the nodes in $J^1$ are closer to $i$ than any of them, and so on. Each of the sets $J^k = \bigcup_{i=1}^{|V|} J_i^k$, where $V$ are the nodes of the graph $G = (V, E)$, is called a **jump**.

**b)** The edges $E_k = \{(i_k, j_k) \in V \times V : i_k, j_k \in J^k\}$ define the **support of the jump** and the **coefficients** $c(i_k, j_k) = g(d(i_k, i_k))$ are given by a function $g(.)$ of the diffusion distances (for example the neg-exponential).

**c)** Then, the **structural filter** $\mathbf{J}^k = (V, E_k, C_k)$ is an edge-attributed graph where the edge attributes are the coefficients $C_k = \{c(i_k, j_k)\}$. Each structural filter is fully learnable

**d)** Each structural filter $\mathbf{J}^k$ with $k = 0, 1, \ldots, K$ feeds a GNN parameterized by $\mathbf{W}^k$ and the resulting embedding $\mathbf{H}^k = \sigma(\mathbf{J}^k \mathbf{X} \mathbf{W}^k)$ is weighted by a learnable parameter $\alpha_k$ subject to $\sum_{k=0}^{K} \alpha_k = 1$. All the weighted embeddings are concatenated and feed a forward network for classification.

