# OpenReview forum: "Node Classification in the Heterophilic Regime via Diffusion-Jump GNNs"
_ICLR.cc/2024/Conference — Submitted to ICLR 2024_

### Official Review · Reviewer_q3r3 · 2023-10-23

**Soundness:** 2 fair
**Presentation:** 1 poor
**Contribution:** 2 fair
**Rating:** 3
**Confidence:** 4

**Summary:**

This paper studies the node classification problem, especially for graphs with low homophily. Overall, the proposed method is to learn a set of filter/propagation matrices, called "Jump" in this paper; then it propagates messages based on every Jump separately and aggregates node representations from Jumps together for node classification. Another contribution of this paper is a newly proposed homophily/heterophily measure called "Structural Heterophily".

**Strengths:**

S1. The performance of the proposed method is good, or at least comparable to the SOTAs. (Table 1 and 2).

S2. The code is released which ensures the reproducibility of this paper.

S3. Figure 8 is clear to understand and I suggest moving Figure 8 to the main content for better illustration.

**Weaknesses:**

W1. The main drawback of this paper is its organization and presentation, which can be improved.

W2. Some statements/claims from this paper seem problematic and not accurate. E.g., in Eq. (2), seems the propagation matrix/Jumps $J^k$ cannot be strictly termed as filters according to the definition from the graph signal processing [1]. Specifically, if they are constructed according to Eqs (2) and (4), they may not share the same set of eigenvectors with the original adjacency matrix.

W3. The novelty of this paper is not outstanding. From a high-level view, the idea of a filter bank (i.e., a set of filters/propagation matrix) is included in many existing works, e.g., [2-5]. Also, the proposed heterophily measure "Structural Heterophily" is related to the existing measure "edge homophily". The difference between the "Structural Heterophily" and edge homophily is that (1) the numerator of the former is the # of different-label connected node pairs, and the numerator of the enumerator of the latter is the # of same-label connected node pairs; (2) the denominator of the former is topology smoothness and the denominator of the latter is the total # of edges.

W4. In my view, some content is not necessary to be included in the paper, and it may distract readers from the main contribution. E.g., the introduction section includes too much technical content, which is not thoroughly explained within the Intro section, and is not helpful for readers to understand the whole story and the core idea. I suggest authors rewrite a part of them with more plain but intuitive language.

[1] Shuman, David I., Sunil K. Narang, Pascal Frossard, Antonio Ortega, and Pierre Vandergheynst. "The emerging field of signal processing on graphs: Extending high-dimensional data analysis to networks and other irregular domains." IEEE signal processing magazine 30, no. 3 (2013): 83-98.

[2] Luan, Sitao, Chenqing Hua, Qincheng Lu, Jiaqi Zhu, Mingde Zhao, Shuyuan Zhang, Xiao-Wen Chang, and Doina Precup. "Revisiting heterophily for graph neural networks." Advances in neural information processing systems 35 (2022): 1362-1375.

[3] Luan, Sitao, Mingde Zhao, Chenqing Hua, Xiao-Wen Chang, and Doina Precup. "Complete the Missing Half: Augmenting Aggregation Filtering with Diversification for Graph Convolutional Networks." In NeurIPS 2022 Workshop: New Frontiers in Graph Learning. 2022.

[4] He, Mingguo, Zhewei Wei, and Hongteng Xu. "Bernnet: Learning arbitrary graph spectral filters via bernstein approximation." Advances in Neural Information Processing Systems 34 (2021): 14239-14251.

[5] Chien, Eli, Jianhao Peng, Pan Li, and Olgica Milenkovic. "Adaptive Universal Generalized PageRank Graph Neural Network." In International Conference on Learning Representations. 2021.

**Questions:**

Q1. In the 2nd paragraph of Section 2, it mentioned "$l^*=\arg\min l^T\Delta l$, where $l*$ is the smoothest labeling of V after propagating l(B) to l(U) through the edges of the graph". I think this is not accurate. Solely based on the context provided in this paragraph, the $l^*$ should be a vector whose entries are all the same.

Q2. In the 4th paragraph of Section 2, it mentioned "We need to infer a hidden graph $G'=(V,E')$. However, the term hidden graph $G'$ is not used a lot and it is not clear how to infer this hidden graph. What is the relationship between the hidden graph $G'$ and the "Jumps"?

Q3. What is the advantage of Structural Heterophily compared with existing homophily/heterophily measures? Also, according to the statement "For $R > 1$ the graph is heterophilic", so only $R=1$ the graph is homophilic? I think the statement is not accurate.

Q4. On the top of Page 4, it mentioned "unstable states u1, u2," and $\bar{A}_1, \bar{A}_2, \dots$. What are those vectors and adjacency matrices? How to obtain them from the context? In the paragraph next, it mentioned "if we relabel the white node, ....., the new Fiedler vector ul leading to the labeling l does no longer induce a sharp step function". It is confusing since the Fiedler vector should only be determined by the graph topology, and not related to the labeling of nodes. In addition, what is the "sharp step function"?

Q5. In Section 3, what is the relationship/difference between $f_{\Theta}$ and $f_{\theta}$?

Q6. In a subsection named "Exploration by Parallel Jumping", the Jumps $J^k$ is presented as a tuple with three elements, but according to the context, the Jumps $J^k$ should be a matrix with the same shape as the adjacency matrix, which is confusing.

Q7. From Section 4.1 we know the nontrivial eigenvectors $\mathbf{U}$ are approximated by $\mathbf{U}=f_\theta(\mathbf{A})$, and obtained by SGD optimizing the Eq. (3). Why not directly set the whole $\mathbf{U}$ matrix as a free parameter and optimize it directly? What is the benefit of parametrizing it with input as $\mathbf{A}$? The design for this part is not clearly explained.

Q8. On page 9, it claimed that "our method needs higher frequency eigenfunctions in order to capture the extreme degree variability of this dataset". Any reference or evidence?

---

> ### Author Response · Authors · 2023-11-14
> **Answer to Reviewer q3r3**
>
> Thank you so much for your constructive and encouraging comments. Below, we address your recommendations. Please do not hesitate to ask more questions during the discussion period.
>
> **W1**. In the paper, we used a couple of figures (Fig. 1 and Fig. 2) to illustrate respectively the concepts of structural heterophily and jumps hierarchy. These two elements summarize the key
> contributions of the approach.
>
> **W2**. The  $\mathbf{J}^k = (V, E_k, C_k)$ are referred to as "filters" by following MixHop [Abu-El-Haija et al. (2019)]. In MixHop, the filters (which are actually powers of the transition matrix) are not learnable. However, we explicitly use $\mathbf{J}^k = (V, E_k, C_k)$ to emphasize that each edge of the so-called "Structural Filter" is endowed with a learnable weight (see Fig.3 where the tridimensional functions illustrate what extent these coefficients define the support of a filter).
>
> In addition, there is a relationship with graph signal processing: we learn the Fourier Transform of the graph. We are not learning the eigenvectors of the adjacency matrix but those of the normalized Laplacian which are equivalent in reverse order to those of the transition matrix.
>
> **W3**. Concerning the novelty, we are aware that the idea of learning a filter bank is in the literature: see for instance MixHop. However, the value of our approach is a technique for learning the shape of these filters, i.e. the coefficients $C_k$. To sum up, note that in this paper the term "Structural Filter" refers to a learnable attributed graph.
>
> Regarding structural heterophily, please see the answer to the Q3.
>
> **W4**. Due to the limitations of space, we decided to introduce the concepts of structural heterophily and Jumps vs Hops to motivate our approach.
>
> **Q1**. Yes, you are right, the correct formula is: $\ell^{\ast}=\arg\min_{\ell\neq\mathbf{0},\ell\perp\mathbf{1}} \ell^T\triangle\ell$, as in the beginning page 4.
>
> **Q2**. The hidden graph $G'= (V,E')$ encodes the ideal/hidden rewiring of the original graph $G=(V,E)$ where the edges $E'$  mostly link homophilic nodes. Since this is an ideal representation, we only show it in Fig. 3 (Embeddings) as a means of visualizing the homophiliation process.
>
> The relationship between this hidden rewiring and the jumps is as follows. Each jump is a graph, i.e. one of the $K + 1$ versions of  $G'= (V,E')$. In this regard, *Diffusion-Jump GNNs* work as a non-linear combination of parallel rewiring. However, we do not originally mention this interpretation for the sake of clarity.
>
> **Q3**. As you noted in W3, structural heterophily is related to edge homophily. However, this relationship is not as straight as you suggest. Edge homophily simply counts the number of homophilic edges, independently of how they are located in the graph. For instance, intra-class heterophilic edges are quite structurally different from those in the bottleneck: the first ones are more difficult to bypass than the natural heterophily arising from the bottleneck. Bottleneck heterophily can not be avoided, but edge heterophily counts these edges and overestimates the difficulty of the problem.
>
> Our proposed metric focuses on those intra-class edges and discards bottleneck edges (see Fig.1 A, where structural heterophily is one). The denominator in our metric is the Fiedler value, whereas the numerator is close to one of the eigenvalues of the Laplacian. As a result, our measure approximates the ratio between two structural variabilities. Which is quite different from counting edges.
>
> **Q4**. To begin with, $\bar{A}$ is a partition, not an adjacency matrix: $\bar{A} \cap A = \emptyset$. Actually, the white node in Fig.2 belongs to $\bar{A}$. Then, the notation $\bar{A}$: $\bar{A}\subseteq \bar{A}_1\subseteq \bar{A}_2\subseteq\ldots$ indicates the level sets depicted in Fig.2 bottom.
>
> Answering your question, $\mathbf{u}_1,\mathbf{u}_2,\ldots$ lead to energies $\mathbf{u}_1^T\triangle\mathbf{u}_1$, $\mathbf{u}_2^T\triangle\mathbf{u}_2,\ldots$
>
> In addition, we are aware that the Fiedler is independent of any labeling. And this is why it is the denominator of our proposed measure. Homophilic graphs tend to have the same energy (numerator) as the Fiedler vector: labeling meets optimal structural partition. In the heterophilic regime, however, it is very likely to have energies (denominator) larger than the Fiedler value.
>
> Concerning the sharpness of the Fiedler vector, it is well known that the smaller the bottleneck, the sharper the partition. But you are right: the sentence "the new Fiedler vector" should be "the new vector" and it is corrected in the new version, thank you for your awareness.
>
> **Q5**. We wanted to discriminate between the parameters of the MLP $f_{\theta}(\mathbf{A})$  and the parameter $\Theta$ of the full model.

---

> ### Author Response · Authors · 2023-11-14
> **Answer to Reviewer q3r3 Part 2**
>
> **Q6**.  Jumps $J^k = (V,E_k)$ are graphs, we should indicate that at the end of the first sentence of the paragraph **Exploration by parallel Jumping**. Filters $\mathbf{J}^k = (V,E_k,C_k)$ are just attributed graphs. Thank you again for making this point more clear in the new version.
>
> **Q7**. In section 4.1, we state that:
> In Appendix B1, we also give practical evidence of the need to solve Trace-Ratio problems in an SGD context, instead of solving the original Trace problem in Eq.2. We also justify the convenience of conditioning $\mathbf{U}$ to $\mathbf{A}$, $\mathbf{U}=f_{\theta}(\mathbf{A})$. Actually, this setting is inspired by how the LINKX method exploits the graph topology.
>
> We clarify this point in a pdf in the supplementary Rebuttal_PDF.pdf
>
> **Q8**. This claim is referred to the fact that we are outperformed by FSGNN using 8 hops. Please, note that our method is shallow. We want to emphasize that certain types of graphs require a higher number of labels in order to obtain a more precise classification with a single layer. The discussion of multiple-layered *Diffusion-Jump GNNs* is in progress.

---

> ### Comment · Reviewer_q3r3 · 2023-11-22
>
> I thank the authors for the detailed rebuttal. I acknowledge your efforts in the rebuttal phase and here is my response and some further suggestions.
> 1. Incorporating review comments into the revised version should be very helpful. In my personal evaluation, the current version is still not ready to be published in one of the top machine learning conferences so I will keep my score.
> 2. Avoid using too much jargon/terms, e.g., "hidden graphs", which are distracting. Try to introduce your core ideas and techniques with minimal but sufficient terms/jargon, in my view. Also, for the term "jump" matrices, I still believe that they cannot be termed as filters. In graph signal processing, filters have a clear definition (an analogy from the classic Fourier analysis). A possibly better term is propagation matrix (if I understand correctly).
> 3. Ask your colleagues/collaborators **who are not familiar with your work** to proofread and check if all the terms/notations are clearly defined and introduced.

---

### Official Review · Reviewer_BSdo · 2023-10-28

**Soundness:** 1 poor
**Presentation:** 1 poor
**Contribution:** 1 poor
**Rating:** 1
**Confidence:** 4

**Summary:**

The paper proposes to learn latent graph connections to deal with heterophilic graph datasets.

**Strengths:**

Difficult to judge due to poor presentation.

**Weaknesses:**

1. The paper is poorly written with important notations not defined and explained. This makes it very difficult to judge the contributions of the paper. Examples include the issues raised in the following questions.

1. The paper seems to be written in a hurry and not proofread. It is not ready for submission.

**Questions:**

1. The abstract is too long and does not capture concisely the contributions of the paper. This is not how an abstract should be written.

1. What is the difference between $f_{\Theta}$ and $f_\theta$? There are notation confusions throughout.

1. In (1), what is $\ell$ and is this the same as $\ell^*$ defined previously?

1. On page 4, from the phrase "Consequently, the jump hierarchy defines a succession of unstable states" onwards, the terms have not been explained clearly. What are "unstable states"? What is an "expansion"?

1. The section "Exploration by Parallel Jumping" needs to be rewritten for clarity. What are "structural" filters? Why are there $K+1$ of them? How do you choose $K$? The authors seem to want to say that they are learning a graph filter but instead tries to bring in confusing jargon.

1. How is the proposed approach different from graph rewiring?

---

> ### Author Response · Authors · 2023-11-21
> **Response to reviewer BSdo**
>
> **General response** . We are deeply discouraged because the review is not very constructive in order to start a discussion. The summary has one line, the paper is judged to be written in a hurry and the decision is "binary" (seems irrevocable). The review seems more a subjective proofreading than a objective scientific judgement.
>
> However, we politely ask the reviewer to engage in a constructive discusion on the ligth of the responses to his/her questions beyond bold format statemets such as "This is not how an abstract should be written".
>
> **W1. Paper poorly written.** What exactly you don't like from the paper? How can we improve it?
>
> **W2. In a hurry.** Well, from the abstract to the supplementary stuff there is a formal construction flowing from intuition to inspiring methods and stopping at the contribution. We ask the following scientific question: *Could a GNN learn the eigenvectors of a graph and use them for node classification?* This is not a minor question (no work in the literature does this: eigenvectors are usually pre-computed). As this is not a minor question it deserves a motivation (measuring heterophily in a different way) and the use of the eigenvectors in a node distance. Having the distance to hand we explore in parallel different level-sets to solve the semi-supervised node-classification task.
>
> **Q1**. Abstract is within the limits of the template. The abstract has 3 stages: motivation (heterophily), technique (related to the state of the art) and optimization problem (trace-ratio) with experimental results.
>
> **Q2**. Difference between $f_{\Theta}$  and $f_{\theta}$. $\Theta$ refers to the whole parameters of the GNN whereas $\theta$ refers to the MLP parameters. What other confusios do you see?, please let us know.
>
> **Q3**. What is $\ell$? $\ell$ is defined in the first paragraph of section 2; it is a labeling. After that, we define $\ell^{\ast}$. as the optimal (minimal energy) labeling.
>
> **Q4**. Jump Hierarchy and expansions (page 4). $\bar{A}$ is a partition of the vertex set: $\bar{A} \cap A = \emptyset$. Actually, the white node in Fig.2 belongs to $\bar{A}$. Then, the notation $\bar{A}$: $\bar{A}\subseteq \bar{A}_1\subseteq \bar{A}_2\subseteq\ldots$ indicates the level sets depicted in Fig.2 bottom. This is a refinement. Then, $\mathbf{u}_1,\mathbf{u}_2,\ldots$ lead to energies $\mathbf{u}_1^T\triangle\mathbf{u}_1$, $\mathbf{u}_2^T\triangle\mathbf{u}_2,\ldots$ and they are unstable because their energies are lower bounded by the optimal one (the ground energy). $\mathbf{u}^T\triangle\mathbf{u}$ defined by the Fiedler vector $\mathbf{u}$.
>
> **Q5**. Structural filters are weighted graphs, i,e, tuples $\mathbf{J}^k=(V,E_k,C_k)$ where $E_k\subseteq E$ and $C_k$ are the learnable coefficients of these edges. By the way, the concept of *structural filter* is introduced in the abstract.
>
> There are $K+1$ filters because the last one is the original graph to capture the full homophilic case (Homophilic branch: see in the last paragraph of section 4),
>
> **Q6**. Actually the filters are rewirings.
>
> We hope that these clarifications engage the reviews in a discussion.

---

> > ### Comment · Reviewer_BSdo · 2023-11-21
> >
> > Since the authors' response is so late, there is not much time left for clarifications, is there? I have been fully prepared to engage in discussions with the authors to clarify many doubts and uncertainties.
> >
> > In summary, the poor presentation of the paper leaves one uncertain about the exact meaning of what is written. I am used to handling mathematically rigorous writings, and I do not like how many things are left up to interpretation in this paper. E.g., the authors say that $\ell$ in (1) is defined in the first paragraph of Section 2. This is not true at all. I see that $\ell(B)$ and $\ell(U)$ are defined in this paragraph. Is the reader supposed to guess that $\ell = \ell(U)$? While this is doable, it immediately makes me less confident after (1) if I am interpreting the paper correctly. The other example is $f_\theta$, which I believe first appears in (2), with no explanation or definition. In a strict mathematical sense, one would have considered $f_\theta$ to be the same function as $f_\Theta$ defined earlier but now with $\Theta=\theta$. But then here what is $\theta$? This is not even defined anywhere before (2)! If you are abusing notation (which is ok for convenience) using similar symbols for the GNN and the MLP, you should at least alert the reader to this.
> >
> > In graph signal processing or general statistical signal processing, a filter refers to a linear transformation (defined on a vector space). In this response, the authors clarify that a filter is a rewiring. Is this a linear transformation in the traditional sense? What is the vector space here?
> >
> > My comments are just examples of the poor presentation and is not exhaustive. Significant revision of this paper is required to make it more readable.

---

> > > ### Author Response · Authors · 2023-11-21
> > > **Comment to reviewer BSdo**
> > >
> > > **Remark** Since the authors' response is so late, there is not much time left for clarifications, is there? I have been fully prepared to engage in discussions with the authors to clarify many doubts and uncertainties.
> > >
> > > Ok lets go.
> > >
> > > **Q1** In summary.... a labeling makes you *less confident*. The parameters also. We have answer to that in the previous answer. Could we please move on?
> > >
> > > **Q2** $\Theta = \theta$? where?
> > >
> > > **Q3** Signal processing, we use the term "structural filter" to be coherent with MixHop in the state-of-the-art. BTW.. Do you agree that we learn the smallest eigevectors of the Laplacian or not? This is fundamental to our approach as you probably have noted.
> > >
> > > Please, more comments are welcome, An example is only a seed.

---

> > > > ### Comment · Reviewer_BSdo · 2023-11-22
> > > >
> > > > It would be good if the authors can clearly summarize the novelty and contributions of this paper. There are efficient ways to compute the Fiedler vector. How does the current method differ from existing approaches, and why is a "learning" approach utilized? What is the advantage of this approach?
> > > >
> > > > It is also difficult to understand the authors' responses above that consist of incomplete sentences. I urge the authors to write professionally to avoid any misunderstandings.
> > > >
> > > > I have read the comments from the other reviewers. In particular, Reviewer q3r3 has similar concerns as mine. As a lot more work is required of this paper, I am keeping my score.

---

> ### Author Response · Authors · 2023-11-22
> **Response**
>
> **Answer 1**, It would be good if the authors can clearly summarize the novelty and contributions of this paper.
>
> We learn the smallest eigenvectors of the Laplacian and estimate diffusion distances for node classification. This is a unique method in the literature: from a point of view of "algorithmic reasoning", we compute eigenvectors using GNNs. By doing so, we bridge spatial and spectral GNNs.
>
> **Answer 2**. There are efficient ways to compute the Fiedler vector.
>
> What methods? Krylov/Power method cannot deal with large graphs. This is why we "learn" them. Please, try to pre-compute these quantities using the current algorithm for a graph of
> 170,000 nodes such as arxiv-year. In addition, these graphs are not connected and a standard algorithm will fail. Moreover, having the learned eigenvectors to hand we can estimate the diffusion distances.
>
>
> **Answer 3** I urge the authors to write professionally to avoid any misunderstandings.
>
> I am professional enough to answer this question.
>
> **Answer 4**
> Reviewer q3r3 has similar concerns as mine. As a lot more work is required of this paper, I am keeping my score.
>
> Thank you. I politely protest against this type of reviews: destructive, uninformative and not helpful at all. Congratulations for your successful blockage. It is a huge damage to the conference.

---

### Official Review · Reviewer_6voN · 2023-11-01

**Soundness:** 4 excellent
**Presentation:** 4 excellent
**Contribution:** 4 excellent
**Rating:** 8
**Confidence:** 5

**Summary:**

This paper addresses the challenges of node classification in heterophilic settings. A key issue identified is the potential for standard GNNs to produce similar embeddings for nodes even if they possess different labels in such regimes. To tackle this, the paper introduces the Diffusion-Jump GNN, a novel model designed to bridge distant homophiles through a "jump" mechanism. This jump is distinct from typical high-order GNNs and is anchored on a structural metric known as the diffusion distance. The paper provides a detailed experiment to validate the model based on multiple datasets and several other different types of models and achieves SOTA on some of the datasets

**Strengths:**

1. The paper provides a comprehensive investigation into the challenges faced by standard GNNs in heterophilic environments, offering a fresh perspective on understanding and quantifying heterophily.
2. The Diffusion-Jump GNN is described in great depth, covering both the mathematical and conceptual foundations. This level of detail facilitates a clear understanding and potential implementation by researchers.
3. By conducting experiments using the same data splits as previous studies, the paper ensures fairness and comparability in the results. Additionally, the provision of hyperparameters and the use of early stopping strategies add reliability to the experiments.

**Weaknesses:**

1. Some mathematical formulas and concepts within the paper may be challenging for non-experts. A more simplified explanation or background could make it more accessible to a broader audience.
2. The Diffusion-Jump GNN might introduce computational complexities, especially related to diffusion distances. This could be a concern in resource-constrained settings or applications requiring real-time responses.
3. Given the complex mathematics and concepts of the Diffusion-Jump GNN, it might not be as intuitive or interpretable as some simpler GNN models. This could impact the acceptance and trust in the model, especially in scenarios where interpretability is crucial.
4. The paper does not delve deeply into the stability of training the model. Some GNN models might be susceptible to issues like vanishing or exploding gradients, which could affect the training efficiency of the Diffusion-Jump GNN.

**Questions:**

1. Given the potential computational complexities, how efficient and scalable is the Diffusion-Jump GNN when dealing with large graphs, e.g., those with millions of nodes and edges?
2. How would the Diffusion-Jump GNN be adjusted or modified when faced with graphs having multiple types of nodes and edges or other intricate structures? Is the method versatile enough for these complex scenarios?
3. Are there plans to further optimize the Diffusion-Jump GNN for better performance or reduced computational demand? Is there potential to extend this method to other graph tasks like graph classification or link prediction?

---

> ### Author Response · Authors · 2023-11-16
> **Answer to Reviewer 6voN**
>
> Thank you so much for appreciating our proposed method and our experimental results. Below we try to respond to your questions. Please, do not hesitate to add more comments during the discussion period.
>
> **W1**. We are aware of that and due to space limitations, we have explained the methods inspiring our approach in the Appendix. We also include some experiments with SBMs.
>
> **W2**. Sure. See answers to Q1 and Q3.
>
> **W3**. We have split the basic maths (in the paper) from the more difficult maths (in the appendix). Regarding the interpretability of the approach for wide use: (1) diffusion distances are approximated by distances between eigenfunctions), (2) our approach can be seen as a learnable directional network where the number of eigenfunctions is the number of classes. However, this number can be changed (reduced or extended) to envision how the diffusion pump captures the structure of the graph. (3) Another source of interpretability is that, in addition, to show how many jumps we need and how important they are, the structural filters become rewirings of the input graph where we show the decisions taken by the GNN to homophiliate the graph (i.e. to construct the latent space. Note the process shown at the bottom of Fig.3).
>
> **W4**. Concerning the stability, it is important to note here that the diffusion pump has a double role: (a) it generates eigenfunctions, and (b) it reduces the structural energies (if needed) of the modifications proposed by the classification loss (regularization). We implement a shallow network where gradients are under control. However, the interference between the Dirichlet loss and the classification loss is important during the initial epochs. Then, the Dirichlet loss converges to a basic set of eigenfunctions, and finally, after the stabilization of the Dirichlet loss, the classification loss fluctuates until convergence. We agree that we have to delve into this point, and we will do that in future papers since we honestly believe that diffusion pumps can be useful modules in other GNNs.
>
> **Q1**. Dealing with huge graphs is still a challenge that new GNN layers or architectures have to deal with today. In our case, creating a dense diffusion matrix forces us to use an $O(n^2)$ memory space, just as a Transformer or HO-GNNs would do. That said, in the paper we have tried to include large graphs to show that our performance is still competitive and we leave the window open so that in future work, we do not need all the distances but a selection of them. Making our method sparse and consuming less memory. Part of this sparsification effort relies on computing the eigenfunctions of the transition matrix (which involves the sparse adjacency in the trace-ratio problem instead of the Laplacian).
>
>
> **Q2**. We believe that when Diffusion-Jump GNN can deal with graphs that have multiple nodes or edges.
> One possible way is to follow the "role2vec" approach (https://arxiv.org/pdf/1802.02896.pdf): compute the transition matrix between types of nodes (for instance) and then rely on this transition matrix. A similar approach can be applied to the line graph of the original graph if there are multiple types of edges. Finally, for a multigraph, we should create a diffusion pump for each particular type of edge.
>
> **Q3**. Yes of course, as we said in Q1, we left the window open so that in the future we can work in a more sparse way and also improve the performance. As to the question of whether we are planning on expanding it for other tasks, the answer is yes, we are currently looking at how these eigenfunctions behave and the distances they generate in link prediction. Since we are observing that they do not react in the same way as in node classification task, because these functions have to capture the nature of the graph, instead of changing it as we do in this work in order to connect distant nodes of the same class. Our preliminary experiments in link prediction show that we can replace the diffusion distance matrix by the concatenation of the matrix of features with the eigenfunctions. This allows us to deal with graphs with $0(10^6)$ edges. What we want to emphasize here is the power of empirical eigenfunctions.
>
> Overall, we are happy to answer more questions during this discussion process.

---

### Meta-Review · Area_Chair_iKwu · 2023-12-11

**Metareview:**

As reviewers BSdo and q3r3 commented, the paper seems to have interesting ideas, but the structure and clarity of the paper needs substantial work. This includes:

(1) **Typos** (just a few examples: the citations are not in brackets and merge with the text due to incorrect usage of citet/citep; Theorem A.1 "Dichilet"; "smoohing" on pg 20)

(2) **Loose notation** (just a few examples: on pg 3, what is the argmin over l(U)? u in eq (1) is undefined until the text later on;  u_k is never formally defined on pg 4; in (2) D_ij seems a scalar, how is a norm being applied to it?)

(3)  General **unclarity** in the exposition about what is used as an "inspiration" for the method, and what is formally a part of the method and/or training procedure. Additionally, the merge of discussions of related concepts and prior work in the prose, with the exposition of the proposed method in the paper, makes it very difficult to precisely understand what is part of the method/training.

This made evaluating the paper quite difficult. As Reviewer q3r3 noted, somewhat related (though not identical) methods involving usage of multiple filter banks have appeared previously. The authors are encouraged to submit a cleaned up version of the paper to a future conference, so the community can more carefully evaluate the new ideas in the manuscript.

**Justification For Why Not Higher Score:**

The overall clarity of the paper was lacking, making it difficult to judge what are the new contributions compared to prior work.

**Justification For Why Not Lower Score:**

N/A

---

### Decision · Program_Chairs · 2024-01-16

Reject